# Female calls promote song learning in male juvenile zebra finches

Linda Bistere [1], Carlos M. Gomez-Guzman[1], Yirong Xiong [2] &
Daniela Vallentin [1] ✉

Social interactions promote vocal learning, but the impact of social feedback on this process and its neural circuitry is not well understood. We studied song imitation in juvenile male zebra finches raised either in the presence or absence of adult females. Juveniles learned songs more accurately with a female present, suggesting her presence improves imitation. When female calls correlated with practice, tutees' songs better resembled the tutor's, hinting toward the possibility that females provide practice-specific vocalizations. Intracellular recordings of HVC projection neurons revealed that a subset of these neurons in both juveniles and adults is sensitive to female calls during listening, suggesting a consistent neural mechanism for processing important vocalizations, regardless of age. However, call-related neural responses during singing were observed only in juveniles. These findings highlight how vocalizations, beyond those of the tutor, influence the neural circuits for vocal learning and production.

Vocal learning is a complex skill exhibited by only a small subset of all vocal species[1,2]. Through observation and imitation of a role model, vocal learners practice their vocalizations to achieve a good approximation of the model[3,4]. This learning process is often influenced by the social environment[5–7], including parental care[8,9]. Previous work in songbirds has mainly investigated vocal learning in the presence of an adult male song model (tutor)[10–16] but the extent to which social factors, specifically vocal interactions with a female bird, a companion that is not able to sing[17], influence song development remains largely unknown.

### Social interactions during vocal learning in zebra finches

Vocal learning in zebra finches (*Taeniopygia guttata*) occurs during a developmental critical period – a limited time interval during which a juvenile tutee learns to reproduce the vocalizations of an adult tutor[18,19]. During this period, the physical presence of a tutor compared to a song playback improves the learning outcome[20], which is likely due to the social interaction between tutee and tutor[6,10,14,16]. Whether the physical presence of another conspecific is sufficient to improve song learning cannot be disentangled in the previously described scenario since the male adult zebra finch will also function as the tutor. In contrast, female zebra finches do not sing and would therefore not provide a model courtship song. A previous study has shown that song-contingent visual feedback from female zebra finches to practicing juveniles results in an increase in the spectral similarity of learned song to tutor song[21]. However, not only visual cues can aid juvenile song learning[22], - females also frequently produce short, innate calls[17,23] which can serve as vocal feedback[24]. It has been demonstrated that female birds produce these calls in response to preferred adult songs[25] indicating that female birds are capable of distinguishing different male courtship songs[26–28] and can have preferences for distinct features of these songs[27,29–32]. Thus, females are well equipped to also provide behaviorally relevant vocalizations to developing males. Therefore, by raising juvenile males in the presence or absence of females and simultaneously tutoring them with a controlled song model, we can quantify the impact of female vocalizations on song learning.

### HVC is involved in song learning

During the critical period, the song learning and production pathway is in a state of increased plasticity[33–35]. Listening to a tutor song is sufficient to induce activity changes in the premotor nucleus HVC (proper name)[34,36,37], a brain area essential for song production in adults[38,39]. Throughout the song-learning period, HVC premotor neurons develop

[1]Max Planck Institute for Biological Intelligence, Seewiesen, Germany. [2]University of Tübingen, Tübingen, Germany. ✉e-mail: daniela.vallentin@bi.mpg.de

a sparse spiking pattern that ultimately drives the production of the learned stereotyped song[33,34,38]. Tutor song-evoked responses in HVC premotor neurons are then suppressed by HVC inhibitory interneurons in adult zebra finches, once learning is complete[34]. In singing adult males this song-related neural activity remains unaffected when females call[40]. Whether vocal responses from females can influence song outcomes by modulating the activity of HVC projection neurons in juvenile birds is not known.

We used behavioral assays to determine if the presence of a non-singing female increases the similarity between a learned song and its model. We recorded the vocalizations from female birds while juveniles were practicing their songs and assessed how these vocal interactions changed during the learning phase. Then, to determine whether female vocal input can impact the underlying song-learning circuitry, we performed intracellular and extracellular recordings in HVC of awake and listening to juvenile and adult birds while presenting playbacks of female calls or playbacks of the bird's own song with overlapping female calls. Last, we also recorded intracellularly in freely moving and singing juvenile birds to decipher whether female calls have an impact on the ongoing activity within this premotor nucleus.

We found that raising juvenile males with non-singing females increased the similarity of learned song to tutor song and was associated with increased female calling behavior as learning progressed. We also found that auditory input from females changed the activity of premotor neurons of HVC. These findings suggest a mechanism of social interactions by which a non-singing instructor can guide developmental song learning.

## Results

### Female presence leads to improved song learning in juvenile male zebra finches

To assess whether the presence of a female zebra finch has an impact on the song-learning outcome of juvenile male birds, we raised juveniles in two different social contexts: alone or with a female bird (Fig. 1A). In both conditions juvenile birds were presented with the same tutor song playback consisting of two synthetized syllables repeated twice (ABAB tutor song, Fig. 1B) which could be elicited by pecking a key. The tutor song playback had a predefined duration and spectral features to which the juvenile song could be compared to (see "Methods" section). Once the birds reached adulthood we quantified the similarity between the learned song and the tutor song (Fig. 1C) using a feature-based approach[41]. We found that songs produced by birds raised with an adult female were more similar to the tutor song (median similarity = 46.17 %, $n = 18$ birds) than songs produced by birds raised without a female (median similarity = 28.44 %, $n = 15$ birds, Fig. 1D).

When analyzing the syllable rate, here defined as the number of syllables (syll) sung per second (excluding the silent gaps) we found that birds raised with a female had a lower syllable rate (with female: median syllable rate = 5.22 syll/s, alone: median syllable rate = 6.88 syll/s, Fig. 1E). Interestingly, birds copied the syllable rate of the tutor song playback (5.87 syll/s) when raised with a female (5.44 ± 1.17 syll/s, $p = 0.137$, one sample $t$-test) but not when raised alone (7.37 ± 1.91 syll/s, $p = 0.0091$, one sample $t$-test). Further, birds raised with a female had longer syllable durations (median syllable duration = 191 ms) than birds raised alone (median syllable duration = 145 ms, Fig. 1F). For both groups the syllable duration was different from the tutor song playback (mean syllable duration = 170 ± 18 ms, with female: $p = 0.045$, alone: $p = 0.0117$, one sample $t$-test). In contrast, the gap duration was not different between groups (with females, mean gap duration = 58 ± 16 ms; alone: mean gap duration = 51 ± 18 ms, $p = 0.1639$, Wilcoxon rank sum test, Fig. 1G) but different from the tutor song playback (mean gap duration = 121 ± 3 ms,). This result is in line with previous studies, showing that zebra finches have a bias toward species-specific gap durations,

which were shorter than the gap duration of the synthesized ABAB tutor song[42,43].

Previous work has demonstrated that the ability to learn specific tutor songs is partly influenced by genetics, and the extent to which the song is inherited depends on how effective the tutoring was[44]. We investigated whether the presence of a female can mitigate genetic biases or influence any father song memory that juveniles may have formed before isolation from their genetic fathers at 30 days posthatch. To address potential confounding factors arising from similarities between the fathers' songs and the tutor's song, we first examined acoustic similarities between these sounds. The analysis revealed that the tutor song exhibited distinct acoustic characteristics different from the fathers' songs (Fig. 2A, B). We then compared the learned songs of a subset of juvenile birds with known father identities ($n = 16$ birds) to the songs of their genetic fathers (Fig. 2A, $n = 11$ father birds). We observed, that birds raised with a female had a higher song similarity to the tutor song and a better match of the tutor's song syllable rate compared to the isolated ones (song similarity: $p = 0.007$, syllable rate: $p = 0.003$, linear mixed-effect model, Fig. 2A–C) independent of their genetic background or early sensory experience prior to 30 days post-hatch.

To investigate any potential influence of the genetic father on song learning, we directly compared the similarity of the pupil's song to both the tutor's song and the genetic father's song. We found that birds raised with a female did not exhibit a significant difference in song similarity to either the tutor or the father (tutor song similarity = 45.91 ± 22.83; father song similarity = 36.61 ± 13.4, $p = 0.2817$, two-sample $t$-test, Fig. 2B). However, birds raised in isolation showed significantly higher similarity to their father's song than to the tutor song (tutor song similarity = 20.41 ± 6.67; father song similarity = 48.34 ± 13.09, $p < 0.001$, two-sample $t$-test, Fig. 2B). Furthermore, the syllable rate of songs learned by birds raised with a female differed significantly from their genetic fathers' syllable rate (pupil = 5.17 ± 0.72 syll/s; father mean = 8.08 ± 3.29 syll/s; $p = 0.014$, two-sample $t$-test,). In contrast, birds raised in isolation did not show a significant difference in syllable rate compared to their genetic fathers (pupil = 8.03 ± 1.8 syll/s; father = 8.52 ± 4.38 syll/s; Fig. 2C).

In summary, our findings suggest that the presence of a female can positively influence song learning in juvenile birds, enhancing similarity to the tutor song and improving the match to the syllable rate of tutor's song, irrespective of genetic background or early sensory experiences. Females might use different sensory modalities to provide behaviorally relevant cues. It has been previously demonstrated that a specific visual display (fluff ups) by females during juvenile song practice leads to a better singing performance[21] but even blindfolded juveniles will produce a more similar copy of the tutor song when a female is present[22]. Additionally, female birds increase their vocalization rate when hearing adult male song[24]. Therefore, we hypothesized that the female calls in relation to the practicing juvenile might also support song learning.

### Female calling behavior increases with improved song performance

To explore the vocal behavior of female zebra finches in relation to juvenile song practice we individually raised six juvenile males with four different adult females while also providing tutor song playback. To identify the vocalizations produced by the females and separate them from the juveniles' vocalizations we equipped the females with small telemetric microphones that were attached to their backs, which predominantly captured sounds produced by the females[45]. Simultaneously, we recorded the sounds produced by both animals with a microphone attached to the cage (Fig. 3A, B). To detect the juveniles' songs and the female calls, but exclude any other sounds that might have been picked up by both microphones, we trained a

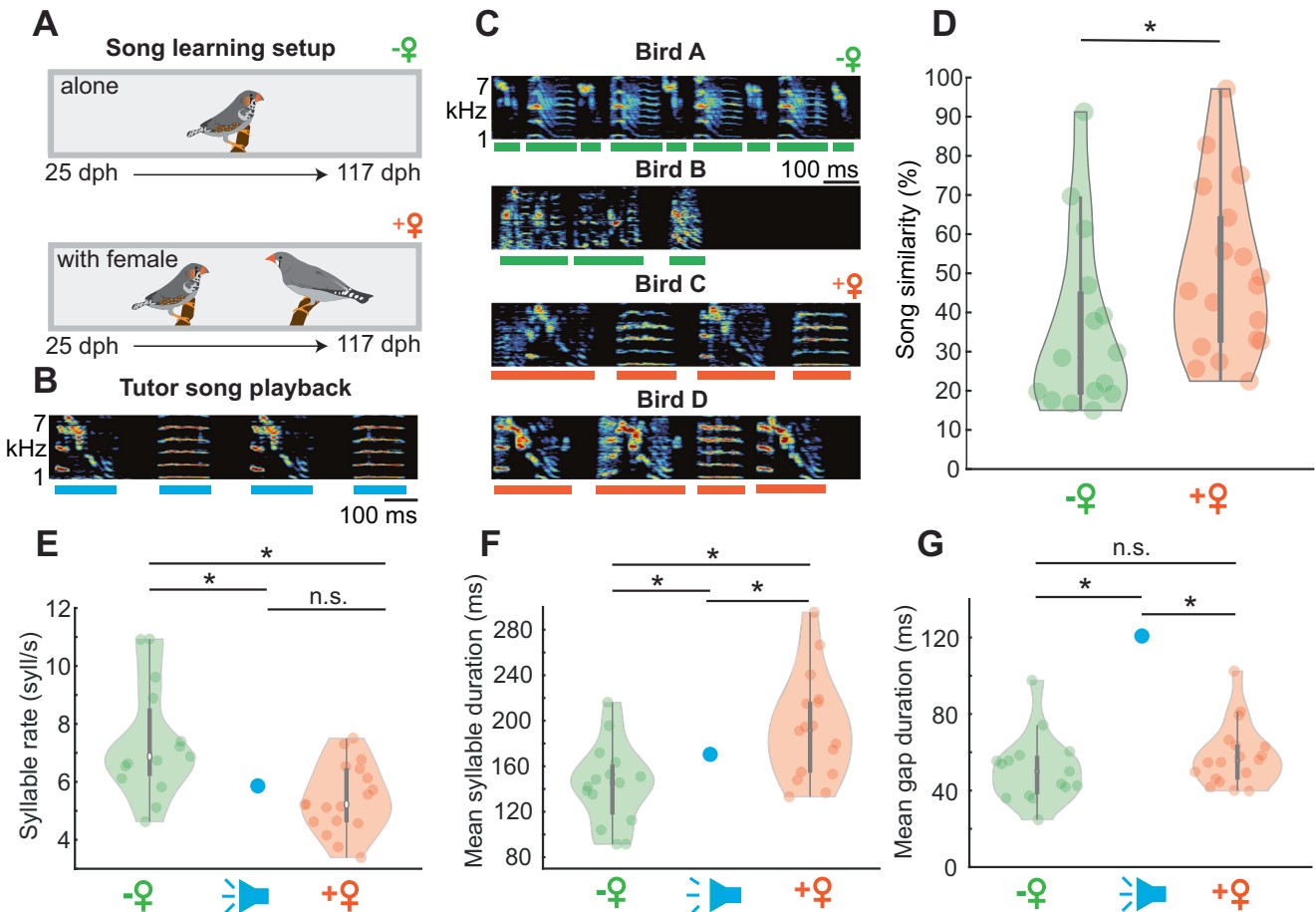

**Fig. 1 | Female presence supports song learning. A** From day 25 to day 117 post-hatch (dph) individual juvenile male zebra finches were trained with a tutor song playback in two social conditions – alone ($n = 15$ juveniles) or with an adult female ($n = 18$ juveniles). **B** Sonogram of the tutor song playback that could be elicited by a packing key ($n = 40$ playbacks/day per bird). Blue lines indicate the syllables. **C** Examples of sonograms of learned song from 4 birds older than 91 dph, green and orange lines indicate syllables. **D** Song similarity of birds raised alone (green, $n = 15$ birds, violin plot: min = 15, max = 96.1, center = 28.44) or raised with a female (orange, $n = 18$ birds, violin plot: min = 22.44, max = 97.11, center = 46.17, Wilcoxon rank sum test, $p = 0.02$). **E** Syllable rate of birds raised alone (green, $n = 15$ birds, violin plot: min = 4.63, max = 10.93, center = 6.88), of the tutor song playback (blue) and of birds raised with a female (orange, $n = 18$ birds, violin plot: min = 3.39, max = 7.51, center = 5.22, Wilcoxon rank sum test, $p = 0.0029$). Asterisk marks

significant difference ($p < 0.05$); n.s. = not significant. **F** Mean syllable durations produced by birds from both groups (birds raised with a female, $n = 18$ birds, violin plot: min = 133.2 ms, max = 295.2 ms, center = 191.5 ms; birds raised alone $n = 15$ birds, violin plot: min = 91.5 ms, max = 216.1 ms, center = 145.4 ms, Wilcoxon rank sum test, $p = 0.0029$). Mean syllable duration of tutor song playback in blue. **G** Mean gap durations per bird of learned song from both groups (with female: $n = 18$ birds, violin plot: min = 39.94 ms, max = 102.3 ms, center = 55.51 ms; alone $n = 15$ birds, violin plot: min = 24.79 ms, max = 97.52 ms, center = 50.02 ms, gap duration with female vs. without female: Wilcoxon rank sum test, $p = 0.1639$, gap duration tutor song vs. with female: one sample $t$-test, $p < 0.001$, gap duration tutor song vs. alone: one sample $t$-test, $p < 0.001$) and tutor song playback. Source data are provided as a Source Data file.

neural network[46] with manually assigned vocalizations as a training set (see "Methods" section). We identified 98,344 juvenile songs (13,572–24,712 songs per juvenile) and 18,672 female calls (932–8232 female calls per juvenile) which were produced during time windows between the juveniles' song onsets and offsets (onset of the first song syllable to offset of the last song syllable). Since the sixth juvenile zebra finch did not learn to produce a regular song but only produced harmonic stack calls (Supplementary Fig. 1), this bird was excluded from further analysis.

To understand whether the females changed their calling behavior in relation to song performance, we assessed the number of female calls emitted during song practice. We observed that, for three out of five juveniles, the number of female calls increased with the age of the juvenile bird (juvenile 1–3: $p < 0.05$, linear regression model, Fig. 3C). Although the female that was initially housed with juvenile 1 later accompanied juvenile 4, we did not observe a correlation between the frequency of female calls and the age of juvenile 4 or

juvenile 5, who was raised with a different female (juvenile 4–5, $p > 0.05$, linear regression model Fig. 3C). Juvenile birds did not produce more songs as the female calling increased throughout the learning phase (Supplementary. Fig. 2). This observation led us to ask whether the female calls might serve as an indicator for the quality of the final song. Therefore, we assessed the similarity of the learned song to the tutor song playback at the end of learning. We found that in the cases when the number of female calls throughout the duration of song (female call ratio) linearly increased with age, the male birds produced songs with higher similarity to the tutor song (song similarity range: 42.33–73.1%, Fig. 3D). Females housed with juvenile 4 and juvenile 5 produced calls that were uncorrelated to juvenile song practice over-development (Fig. 3C). In these cases, calling did not lead to the production of a similar tutor song (song similarity range: 19.44–29.11% Fig. 3D). Although all birds were trained with the same tutor song, only juveniles 1–3 had a higher similarity to tutor song than untutored birds (Fig. 3D, similarity of untutored birds ($n = 7$ birds) to

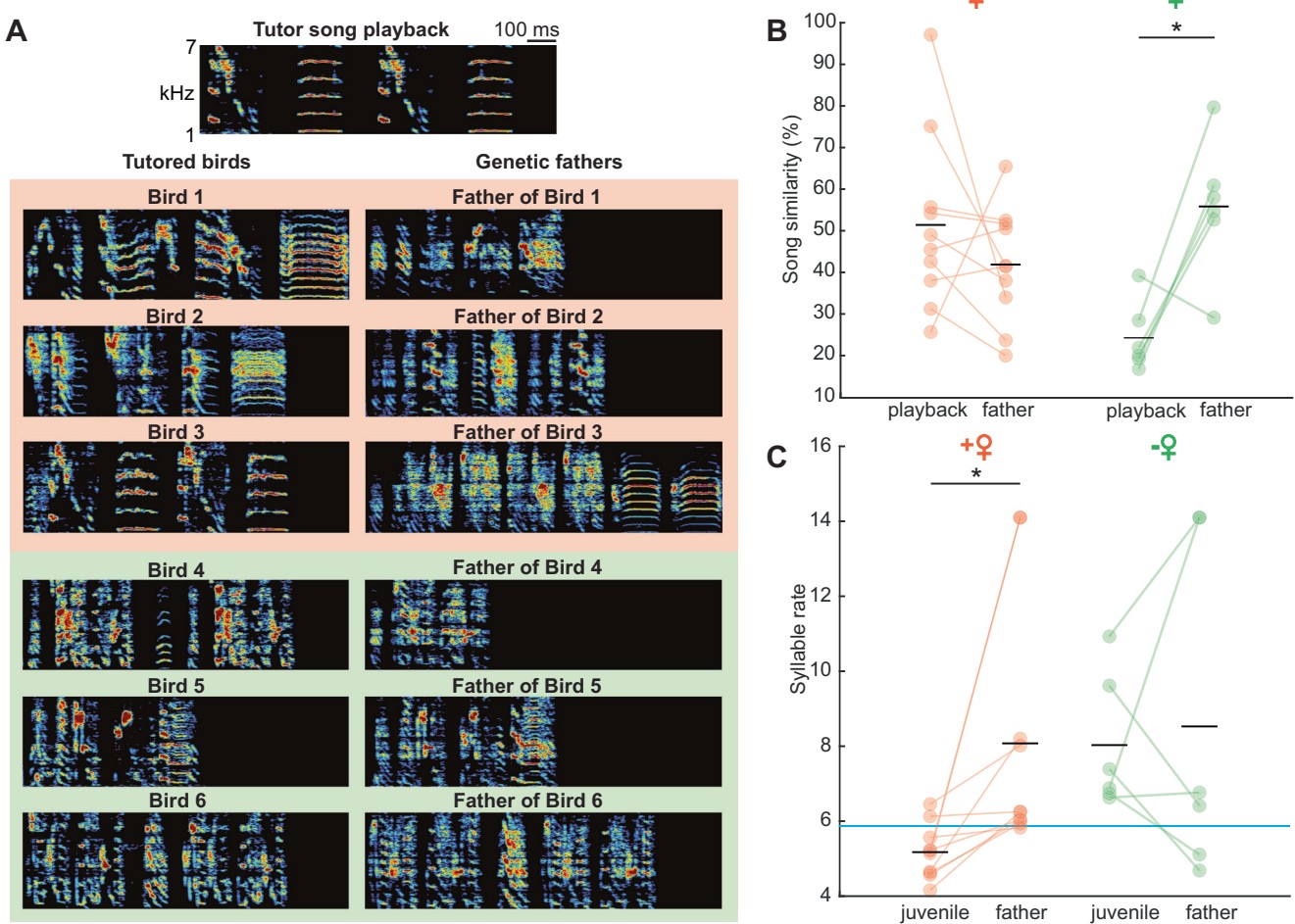

**Fig. 2 | Female presence aids song learning from playback. A** Examples of learned song from tutored birds (*n* = 16 birds) and their respective fathers (*n* = 11 birds). *Top*: ABAB tutor song playback. *Below (left)*: Sonograms of example songs produced by Birds 1–3 raised in female presence and birds 4–6 raised alone *Below (right)*: Sonograms of example songs produced by the genetic fathers of birds 1–6. **B** Song similarity between ABAB playback and the song of the ABAB tutored bird or between the fathers' song and the song of the ABAB tutored bird, from birds raised in the female presence (orange, *n* = 10 birds) and birds raised alone (green, *n* = 6 birds). The tutor song was acoustically distinct from the father's song (similarity (father/tutor) = 31.45 %, similarity (untutored birds/tutor) = 32.02 %, *p* = 0.833, Wilcoxon rank sum test). Asterisk marks a significant difference (*p* < 0.05). **C** Syllable rate comparison between learned song and fathers' song from birds raised with a female (orange, *n* = 10 birds) and birds raised alone (green, *n* = 6 birds, two-sample *t*-test, *p* = 0.80). Blue line represents the syllable rate of tutor song playback. Source data are provided as a Source Data file.

ABAB tutor song playback = 32.023 ± 4.44%). We employed the similarity between untutored birds' songs and the tutor song playback, along with a 95% confidence interval as a threshold, to categorize birds into good and poor learners based on their ability to replicate the provided tutor song. Scrutinizing this criterion (40.7% similarity threshold) led to the observation that juvenile birds, which received progressively more auditory input from females during their song performance, exhibited better replication of the tutor song compared to juveniles that did not experience an increase in calling by females over time (Fig. 3D). When calculating the syllable rate for all five juvenile birds we observed that juveniles 1–3 were closer to the syllable rate of the tutor song playback than juveniles 4 and 5, that received uncorrelated vocalizations (Fig. 3E).

These findings indicate that female birds produce vocalizations in relation to song practice and that this vocal input might lead to a better spectral and temporal copy of the learned song to tutor song.

### Female calls evoke neural responses in HVC projection neurons of listening juvenile and adult males

Perception of the tutor song induces changes in the neural circuitry of the premotor nucleus HVC in juvenile birds[34,36]. Since the premotor nucleus HVC receives auditory inputs[47–49] and is involved in song learning and production[35,50], we asked if female vocalizations can similarly evoke neural responses in the premotor circuitry in juvenile male zebra finches.

We performed intracellular recordings in HVC of awake, listening male juveniles while presenting female call playbacks (Fig. 4A, B). In total, we recorded four antidromically identified HVC$_{RA}$ neurons (neurons that project to the robust nucleus of the arcopallium (RA) and are directly involved in song production) and 19 unidentified HVC projection neurons. Since no quantifiable difference in neuronal response patterns could be observed between the identified and unidentified HVC projection neurons, we pooled the data and considered the recorded neurons as HVC projection neurons (*n* = 23 HVC projection neurons, *n* = 9 juvenile birds). We observed an increase in firing rate during the playback of a female call (silence: firing rate = 3.64 ± 5.02 Hz, female call: firing rate = 6.64 ± 8.16 Hz, Fig. 4C). Next, we analyzed the spiking precision which describes how stereotyped and time-locked spiking events are related to a female call. Due to the variability in the firing properties of each neuron, we initially computed spiking precision during periods of silence as a baseline. Subsequently, we compared the baseline spiking precision to the spiking precision

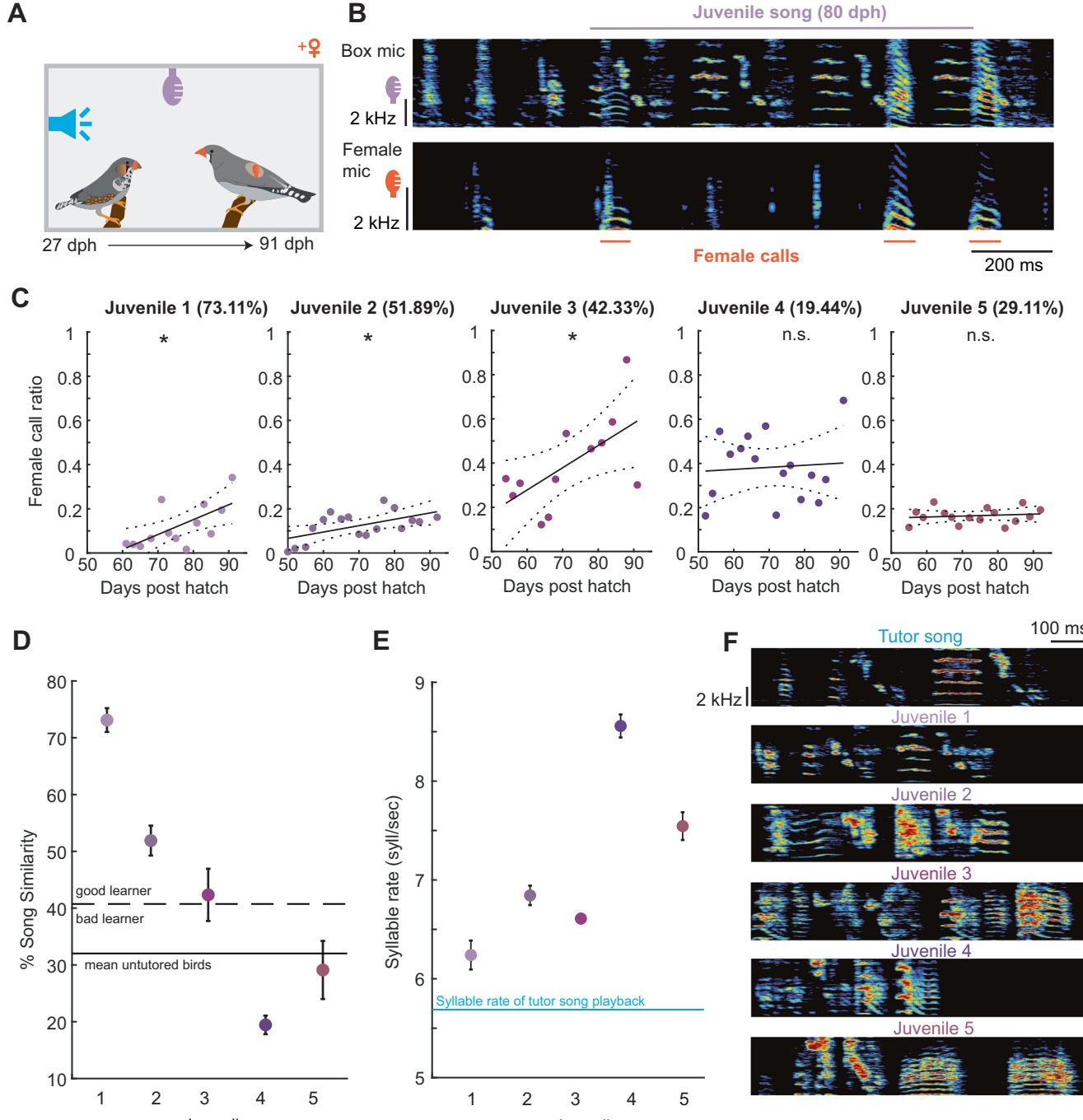

**Fig. 3 | Correlated female calls to juvenile practice improve song learning.**
**A** Experimental setup with two microphones – box microphone (purple) and backpack microphone (orange). **B** Example spectrograms of a simultaneous recording from the box microphone and the backpack microphone during song practice. Song (purple) and female calls (orange) were automatically detected using the Deep Audio Segmenter (DAS)[46]. **C** Ratio of female calls occurring during juvenile song practice relative to the duration of the song production (see "Methods" section) for each individual juvenile 1–5 ($n = 5$ juveniles, $n = 4$ different females, linear regression model, juvenile 1: $p = 0.0142$, juvenile 2: $p = 0.0168$, juvenile 3: $p = 0.0329$, juvenile 4: $p = 0.785$, juvenile 5: $p = 0.598$). Purple circles represent the call rate during songs. Black line indicates the linear correlation (juvenile 1: $R^2 = 0.435$, $F = 8.46$, juvenile 2: $R^2 = 0.325$, $F = 7.23$, juvenile 3: $R^2 = 0.38$, $F = 6.12$, juvenile 4: $R^2 = 0.005$, $F = 0.008$, juvenile 5: $R^2 = 0.02$, $F = 0.29$), dotted line shows confidence intervals of calculated correlation. Numbers in brackets indicate the mean song similarity score to the tutor song on the last day of practice. Asterisk marks significant difference ($p < 0.05$); n.s. = not significant. **D** Song similarity (mean value in purple, error bars represent standard error of the mean, $n = 10$ song motifs per juvenile) at day 91 (juvenile 1 and 4) or day 92 (juvenile 2, 3, and 5) post-hatch for all five juveniles. Black line indicates the song similarity of untutored birds, dashed lines indicate the upper bound of the 95% confidence interval calculated from the song similarity of untutored birds. **E** Syllable rate of the song of the individual juveniles ($n = 5$ birds). Error bars represent the standard error of the mean. Syllable rate of the tutor song playback is indicated with the blue line. **F** (*Top panel*) Spectrogram of the tutor song. (*Bottom three panels*) Spectrogram of the learned song of juveniles 1–5. Source data are provided as a Source Data file.

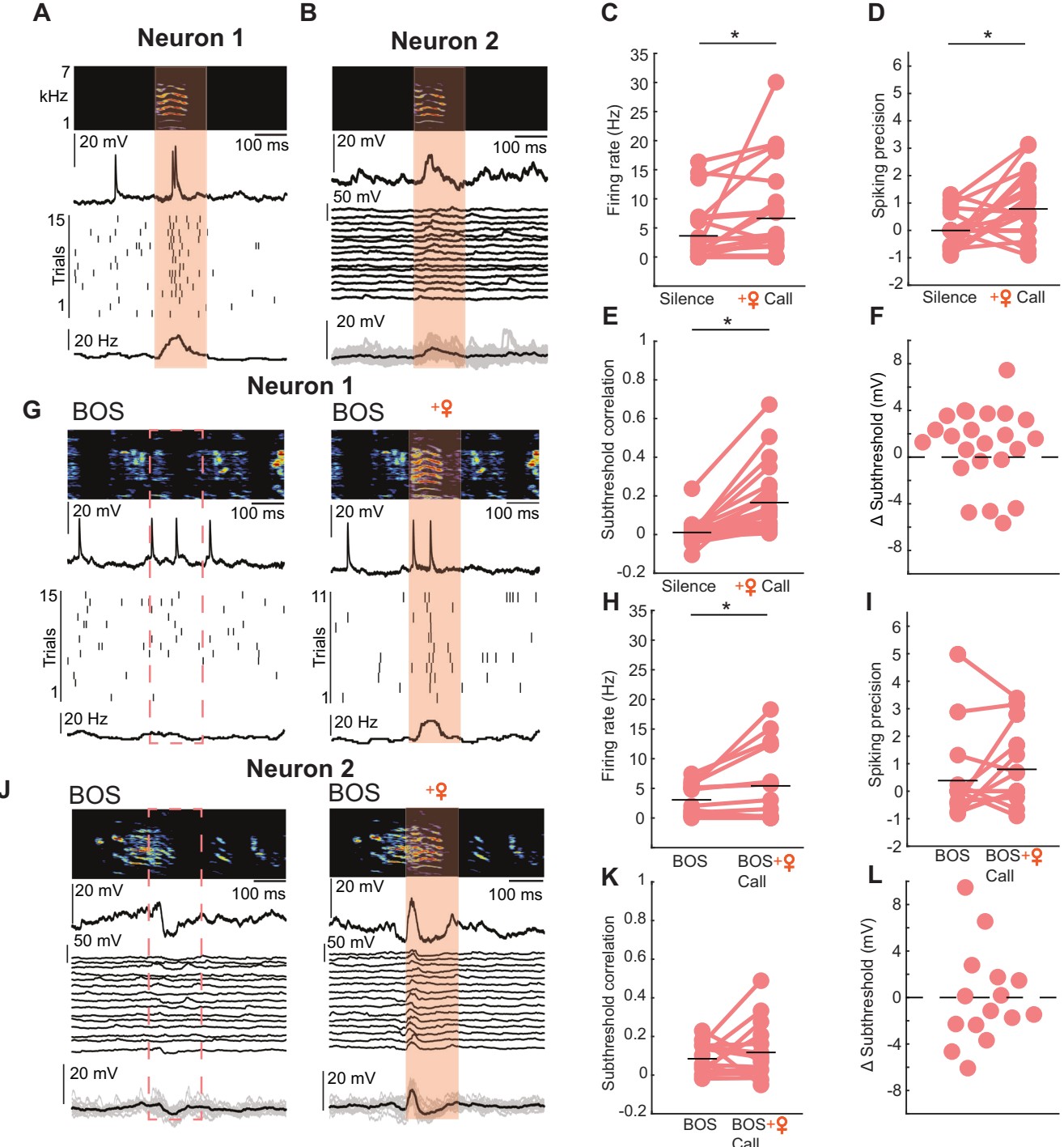

**Fig. 4 | Female vocalizations elicit neural responses in a subset of HVC projection neurons in juvenile birds.** **A** Example recording of an HVC projection neuron increasing spiking activity during female call playback (call onset to call offset +50 ms). The orange-shaded area represents timing of a female call + 50 ms, top: Spectrogram of female call playback. middle: Example trace of the membrane potential of an HVC projection neuron during female call presentation, below: Spike dot raster of the neuron shown above during 15 representations of a female call, bottom: peri-stimulus time histogram (black line). **B** Recording of an HVC projection neuron during 15 representations of a female call. Top: Spectrogram of female call playback, middle: example trace of membrane potential, below: membrane potential traces from 15 trials, bottom: average membrane potential (black) and trial membrane potentials overlaid (gray). **C** Firing rate of all recorded cells during silence and call playbacks (*n* = 23 projection neurons recorded from 9 birds, linear mixed-effect model, *p* = 0.0286) black lines indicate the mean. Asterisk marks a significant difference (i < 0.05). **D** Spiking precision of 23 recorded cells (linear

mixed-effect model, *p* = 0.005). **E** Subthreshold correlation of all 23 cells (linear mixed-effect model, *p* < 0.001). **F** Difference (Δ) of subthreshold activity between silence and female call playbacks. **G** Example recording of an HVC projection neuron (same cell as in **A**) with increasing spiking activity locked to female call during bird's own song playbacks. Left: Segment of BOS without female call, right: same segment with female call. **H** Firing rate of all 15 cells during the segment of BOS with the overlapping female call and during the same segment of BOS without a female call (linear mixed-effect model, *p* = 0.03). **I** Spiking precision comparison as in (**H**). **J** HVC projection cell (same cell as in **B**) with subthreshold responses to female call playback within BOS. **K** Subthreshold correlation comparison as in (**H**). BOS: subthreshold correlation = 0.08 ± 0.08, BOS + female call: subthreshold correlation = 0.12 ± 0.15, p = 0. 405, linear mixed-effect model. **L** Difference (Δ) in membrane potential (as in **F**) between BOS playback and BOS playback interleaved with the female call (Δ subthreshold = −0.07 ± 4.09 mV, *p* = 0.94555, linear mixed-effect model). Source data are provided as a Source Data file.

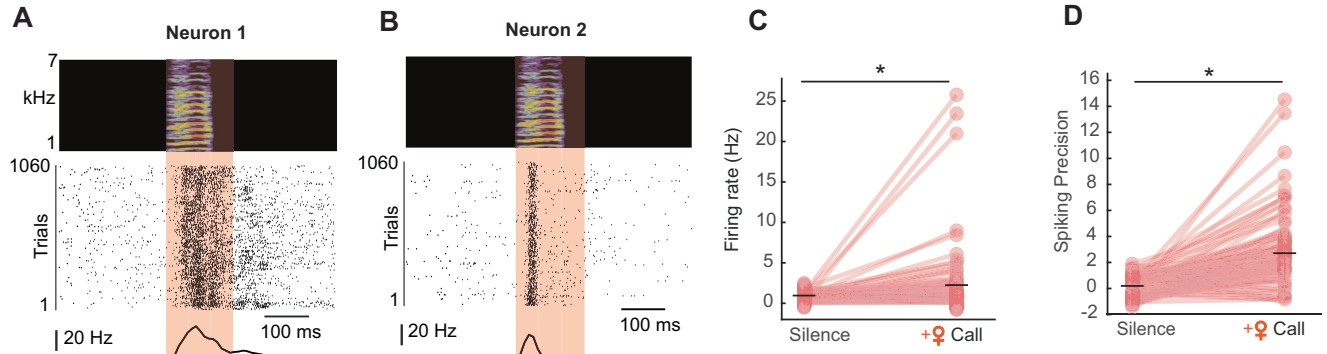

**Fig. 5 | Female vocalizations elicit neural responses in a subset of HVC projection neurons in adult birds. A** Example HVC projection neuron (neuron 1) extracellularly recorded during female call playbacks (call onset to call offset + 50 ms[67]). Top: sonogram of female call playback, middle: spike dot raster, bottom: peri-stimulus time histogram. **B** Neuron 2, as in (**A**). **C** Firing rate of 133 HVC projection neurons recorded during silence and during the female call playback (linear mixed-effect model, $p = 0.001$). Asterisk marks a significant difference ($p < 0.05$). **D** Spiking precision of 124 HVC projection neurons recorded during silence and the female call playback (linear mixed-effect model, $p < 0.001$). Source data are provided as a Source Data file.

observed during female calls. We found that HVC projection neurons exhibit precise spiking events in response to female calls (silence: spiking precision = $-0.003 \pm 0.62$, female call: spiking precision = $0.78 \pm 1.14$, Fig. 4D). This observation led us to ask whether HVC projection neurons receive time-locked, stereotyped inputs during female call presentation, which might not be reflected when solely analyzing the spiking precision. Therefore, we calculated the subthreshold correlation and found that fluctuations in cell membrane potential were more correlated during a female call compared to a silent period (silence: subthreshold correlation = $0.011 \pm 0.06$, female call: subthreshold correlation = $0.16 \pm 0.17$, Fig. 4E). The overall change in subthreshold input during female call presentation was not different from regular occurring subthreshold activity changes during silence ($\Delta$ subthreshold = $0.9722 \pm 3.275$ mV, $p = 0.160$, linear mixed-effect model, Fig. 4F) indicating that the more correlated activity observed in the subthreshold activity (Fig. 4E) is not systematic but either reflects a release from local inhibition or the transmission of excitatory female call-related information from upstream auditory areas.

Next, we asked if the timing of the female calls with respect to the juveniles' song has an impact on the neuronal activity. Therefore, we presented the juveniles with playbacks of the birds own song (BOS) or with BOS interleaved with a female call while recording 15 of the aforementioned recorded neurons in 6 birds (Fig. 4G). The neurons increased their firing rate in relation to BOS + female call compared to BOS alone (BOS: firing rate = $2.81 \pm 2.94$ Hz, BOS + female call: firing rate = $5.41 \pm 6.31$ Hz, Fig. 4H). In contrast, HVC neurons did not exhibit time-locked spiking activity (BOS: spiking precision = $0.39 \pm 1.58$, BOS + female call: spiking precision = $0.795 \pm 1.38$, $p = 0.205$, linear mixed-effect model). In the BOS + female call condition, neither the subthreshold correlation nor the change in subthreshold activity were significantly different on a population level (Fig. 4K, L). These findings suggest that auditory responses to female calls in HVC are less stereotyped during female calls within BOS playback when compared to auditory responses to female calls alone (Fig. 4E, K).

Next, we asked whether the same response pattern evoked by a female call can be observed in adult birds that cannot undergo further learning[51]. We performed multi-channel extracellular recordings in HVC of awake-listening adult male birds ($n = 5$ birds) that were presented with female calls. Based on firing rate characteristics[52,53] (see "Methods" section), we identified 133 HVC projection neurons. On a population level, HVC projection neurons significantly increased their firing rate in response to female call playback (firing rate during female call = $1.93 \pm 3.6$ Hz, firing rate during silence = $0.92 \pm 0.53$ Hz;) (Fig. 5A–C). Additionally, we also found that HVC projection neurons in adults displayed precise spiking responses to the female calls on a

population level (spiking precision during playback = $2.49 \pm 2.52$, spiking precision during silence = $0.04 \pm 0.89$;) (Fig. 5A, B, D).

This finding indicates that female calls can induce spiking activity in HVC projection neurons outside the context of singing beyond development. Female calls are a salient and positive signal for adults resulting in dopamine release in other brain areas[54]. Our result further supports the notion that females may call to evaluate the courtship performance of the males. During adult song production, however, female calls do not change the stereotyped activity pattern of HVC projection neurons[40]. A previous finding that HVC inhibitory interneurons exhibited a variety of responses to call playbacks[55] suggests that HVC inhibitory interneurons might gate off the auditory information in order to shield HVC projection neurons from undergoing plasticity which might affect the stereotyped production of the adult song.

## Female calls induce responses in neural activity in a subset of HVC neurons in singing juveniles but not adults

The more stereotyped an adult male song is across renditions, the more attractive it is for a female zebra finch[26]. Female calls heard during courtship song production do not change the activity of HVC projection neurons in adult zebra finches[40]. Thus, female calls are unlikely to affect the stereotypy of ongoing male songs, which would presumably be beneficial in the context of courtship. We have presented evidence that vocalizations from the female appear to modulate the outcome of the juvenile song-learning process. Therefore, we asked if the female calls have an impact on HVC projection neuron activity while the juvenile birds are practicing.

We performed intracellular recordings of HVC projection neurons in freely moving, singing juvenile birds ($n = 9$ neurons in 4 birds, Fig. 6A, B). The observed activity profile in juvenile birds older than 76 days post-hatch was similarly stereotyped as the pattern observed in HVC projection neurons of singing adults[40] (juveniles: median subthreshold correlation = 0.81, adults: median subthreshold correlation = 0.80, Fig. 6C). To explore if the neural activity of the projection neurons is affected by the female call, we presented a female call playback during 22 instances of 44 song motifs. On a population level, the neurons exhibited a similar number of spikes during perturbed and unperturbed syllables (#spikes(perturbed) = $0.82 \pm 0.1.59$, number of spikes(unperturbed) = $0.78 \pm 1.36$, $p = 0.917$, linear mixed-effect model, Fig. 6D, F). However, upon analyzing individual neuron activity, we observed that in each juvenile bird ($n = 4$ birds), neurons exhibited altered firing patterns during singing when a female vocalization occurred ($n = 4$ out of 9 recorded neurons). To determine the significance of this change, we compared our data with a previously

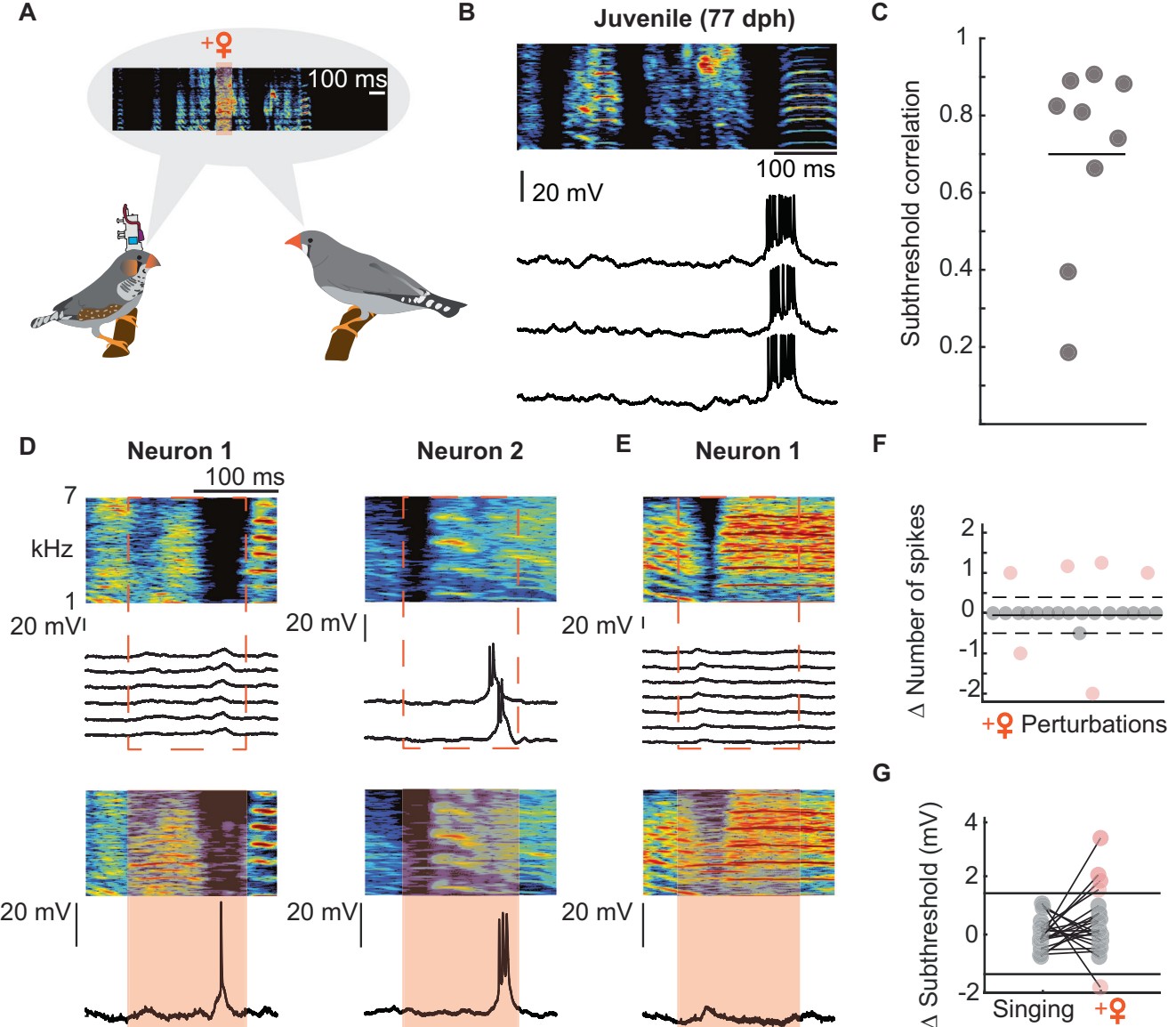

**Fig. 6 | Female vocalizations during juvenile song production change neural activity within a subset of HVC projection neurons. A** Experimental setup for intracellular recordings in freely moving, singing juvenile birds ($n$ = 4 juvenile birds). **B** HVC projection neurons ($n$ = 9 neurons) in juvenile birds older than 76 dph showed stereotyped neural activity during song production. Top: sonogram of the produced song motif. Bottom: intracellular membrane potential of the same neuron recorded during three renditions of the song motif. **C** Subthreshold correlations across song renditions for the recorded HVC projection neurons. **D, E** Examples of female call perturbations during song practice. Top: Spectrogram of song segments with corresponding membrane potentials for each cell during song production. Dashed line represents when a female call will occur on the next song rendition. Bottom: corresponding song segments to the top, during female call perturbations. Below: Membrane potential during call perturbation. Orange-shaded area highlights the time from call onset to call offset +50 ms. **F** Delta (Δ) of the number of spikes during female call perturbations ($n$ = 22 perturbations in nine HVC projection neurons). Black line indicates |mean Δ number of spikes| during singing in adult zebra finches (data from Vallentin and Long, 2015[40]), dashed line indicates a confidence interval of 1.96 standard deviations from the mean Δ number of spikes during juvenile's singing. Orange dots highlight instances when the number of spikes changed beyond the confidence interval. **G** Δ subthreshold changes during a female call ($n$ = 22 perturbations, $n$ = 9 neurons, $n$ = 4 juvenile birds). Black lines indicate maximally observed Δ subthreshold changes in adult birds (data from Vallentin and Long [40]). Orange dots highlight instances when the Δ subthreshold changed beyond changes observed in adults. Source data are provided as a Source Data file.

published dataset from HVC cells ($n$ = 12 HVC projection neurons) recorded in four adult zebra finches[40]. In contrast to juveniles, the firing rate of projection neurons in adults remained unchanged when females vocalized during song production (Δ number of spikes = 0 ± 0; 0 out of 98 instances of female vocalization during song; $n$ = 12 neurons in 4 adults). This difference was statistically significant to our juvenile dataset (Δ number of spikes = 0.04 ± 0.70, $p$ = 0.0002, Fisher's exact test, Fig. 6D–F). Specifically, in juveniles, a change in the number of spikes compared to baseline firing during song production was

observed in 6 out of 22 cases, whereas no change was observed in any cases in adults (Fig. 6D–F).

To further compare subthreshold changes caused by female call perturbations in adult and juvenile HVC projection neurons, we used the maximum delta subthreshold (as shown in Fig. 3 of Vallentin and Long [40]) during a female's call while an adult zebra finch was singing as a threshold to determine if any activity changes occurred (Fig. 6G). We found that in five out of the 22 cases (observed across five neurons), neurons exhibited subthreshold changes in response to female calls

that extended beyond the maximally occurring changes observed in adult birds. These findings indicate that female call perturbations during song practice in juveniles had an effect on the ongoing membrane potential activity and this might be a potential mechanism by which female calls can induce plasticity within HVC projection neurons. However, when testing whether female calls have a direct effect on the ongoing song performance we did not find a significant improvement of the syllable following female call perturbation (Supplementary Fig. 3). Similar to proposed endogenous mechanisms[56], exogenously introduced variability within this song production pathway could potentially facilitate exploratory singing behavior over subsequent song motifs, until a more accurate song copy has been achieved.

## Discussion

Social cues are important for learning[7] but measuring their exact contribution is often difficult. By raising juvenile zebra finches with a tutor song playback accompanied by a female bird or alone, we were able to quantify the impact of female calls on the song-learning performance. We found that the female presence during the learning process of juveniles improved tutor song copying (Figs. 1 and 2). Furthermore, in juveniles that copied the tutor song more accurately, the female calling rate increased along with the progression of the song-learning phase (Fig. 3). Correspondingly, female calls heard during the song development period resulted in activity changes in HVC projection neurons (Figs. 4 and 6). This female call-evoked activity persisted during adulthood (Fig. 5), outside of the context of song production, and might serve as a general mechanism for signaling the quality of a song or encouraging song production.

### Song learning in differentially socialized juvenile zebra finches

The presence of a live tutor during learning enhances the ultimate similarity of the learned song to the tutor song, compared to hearing tutor song playbacks alone[20]. However, in these scenarios, it is difficult to identify the social factors aside from the song that together influence learning. We therefore paired juveniles with adult non-singing females to add a social component without allowing for tutor-tutee interactions. Since females have preferences for certain features of male song[26] and these preferences largely depend on early exposure[25,27] it is plausible that adult females also apply their learned song preferences to influence and shape juvenile song learning.

The mere presence of a sibling female has been shown to enhance the copy to tutor song in blindfolded juvenile zebra finches[22]. It is therefore plausible, that even in the absence of visual cues, male juvenile birds are supported throughout the song-learning process by females. Additionally, we noted that birds raised in the presence of females were less inclined to learn a song similar to their father's song. Early life exposure to fathers' songs can already be sufficient for male juveniles to learn the fathers' song[57,58]. However, this early life experience can be overruled, if the juvenile birds are tutored by a male tutor[44]. In our study, we show that the presence of a non-singing female is sufficient to copy the tutor's song rather than the father's song.

An additional possibility of how females might promote song learning is their interactions with the tutor song which might be sufficient to enhance sensory acquisition in juvenile birds. We were not able to rule out that females display preference-related behavior when hearing the tutor song playback. However, social interactions with the pupil play a major role in how juveniles select tutors and ultimately in their song-learning success[6]. We, therefore, hypothesize that direct social interactions with a female bird would have a greater influence on juvenile song learning than juveniles' passive observation of female preference behavior toward a song playback.

### Female calls could guide song practice

Since females do not sing but only produce calls[25], calling during song practice is a candidate mechanism with which females can direct juveniles to adjust their song performance during the critical period. Our results demonstrate that females produce calls during song practice and increase their call rate as song learning progresses. When female vocalizations were not correlated with the song practice during the critical period for vocal learning, the learned song was least similar to the tutor song. Throughout development, juvenile birds produce songs that vary greatly in their spectral and temporal features[59]. Female birds produced more calls as the juvenile songs became more stereotyped, raising the possibility that female calls might function as rewarding stimuli when juvenile birds produce a preferred version of the song. Although it has been shown that the female presence alone does not trigger dopamine release in HVC of juvenile birds[60], female calls are perceived as rewarding stimuli in adult birds[54]. Further studies addressing the involvement of the dopaminergic system are thus important to characterize the significance of female vocalizations in learning juveniles. Since our dataset does not provide a conclusive answer to whether female calls alone aid song learning, it is an intriguing future idea to investigate song learning in juveniles who are solely exposed to female call playbacks in a contingent or non-contingent manner in relation to their song practice. Similar to tutor song playback experiments lacking a social partner[20] we would expect that female call playbacks still improve song learning but are less impactful compared to the presence of a live female calling during song practice. Given the potential of female calls to serve as a rewarding stimulus, it should be further explored whether and how female calls also shape the sensory learning of the tutor song.

### Female calls elicit auditory responses in the premotor nucleus HVC of zebra finches

During the learning phase, the premotor nucleus HVC receives auditory input which is later suppressed by synaptic inhibition[34]. Therefore, auditory-evoked activity in HVC projection neurons is absent during song production in adults[34,40]. Our results show auditory-evoked activity in a subset of HVC projection neurons of juvenile birds, in response to female calls. Auditory responses within the song network might support the stabilization of synaptic connections within HVC and therefore promote the production of a preferred and stereotyped version of the song. Female calls also elicit auditory-evoked responses in HVC of listening adult zebra finches outside of the context of song production supporting the notion that this type of vocalization might play a role in directing HVC circuit dynamics.

### Projection neurons in HVC show responses to female calls during song production

Although female calls activate dopaminergic neurons in adult birds[54], auditory responses to female calls in HVC projection neurons appear to be absent during song production in adults[40]. Our observed auditory-evoked responses to heard calls in listening juvenile birds led us to explore if, similar to HVC premotor neuron responses to tutor song, responses to female calls during song production are initially present in juveniles. We observed exemplary changes in neural activity during song production when the song was perturbed by a female call. Although HVC activity was not significantly altered at the population level, we identified a subset of neurons sensitive to female vocalizations during singing in juvenile birds. This sensitivity of HVC neurons may be dependent on the age or developmental stage of the juvenile birds since HVC neurons in adult birds do not demonstrate auditory responses to female calls during song. Our results highlight the importance of social responses from non-singing females during the song-learning phase as well as its potential to shape the neural circuitry for song learning in juveniles. We provide insights into an exogenous

mechanism in learning zebra finches and highlight the contribution of calls, that might deepen our understanding of reinforcement learning within a complex, socially-embedded behavior.

## Methods

### Animal housing

All procedures described were approved by the Regierungspräsidium Oberbayern (VET 02-21-201, VET 02-21-102) or Landesamt für Gesundheit und Soziales (LAGeSo Berlin) (G 0225/16) at the Freie Universität Berlin. Animals were housed with a light cycle from 7 a.m. to 7 p.m. and were provided with food, water, and grit *ad libitum*.

For the song-learning experiments, juvenile male birds (25–35 days post-hatch) were placed in sound-attenuated chambers. Each juvenile bird was housed either with an adult female or alone until the end of the song-learning phase (max. 117 days post-hatch).

For the electrophysiological recording experiments, juvenile birds were raised by their biological parents in mixed-sex aviary or breeding cages. Adult male birds (at least 100 days post-hatch) were housed with female adult birds for at least 72 h before extracellular recordings.

### Song learning

Juvenile birds ($n = 33$ birds) were raised with both parents until nutritional independence (25–35 days post-hatch). Afterward, juveniles were housed in sound-attenuated chambers during the critical period of song learning. Of all 33 juvenile birds, 22 birds were not related to each other. The remaining 11 birds came from 4 breeding pairs (pair A to pair D). Siblings from pairs A–C were distributed in both social conditions – raised in female presence or alone. Both siblings from pair D were raised in the alone condition. In the social context (with females), each juvenile bird was accompanied by one adult female throughout the song-learning phase ($n = 18$ birds). Adult female birds were acquired from the previously described mixed-sex aviaries. In the socially isolated context (alone) each juvenile bird was housed alone ($n = 15$ birds). Both groups were trained as follows: At the beginning of the training phase (first two weeks) birds received 20 passive tutor song playbacks per day (7 a.m. to 2 p.m.). During the following phase, the birds could actively elicit playbacks of the tutor song themselves via pecking keys ($n = 40$ playbacks per day, 7 a.m. to 7 p.m.). The tutor song playback consisted of two synthetized zebra finch syllables repeated twice – as previously described in ref. 61. The first syllable of the pair (syllable A, pitch = 2456.6 Hz, wiener entropy = −3.06; continuity of frequency = 360.53 Hz) had a duration of 185 ms and the second one (syllable B, pitch = 1165.4 Hz, wiener entropy = −3.754, continuity of frequency = 241.41 Hz) had a duration of 155 ms. The silent gap between syllables had a duration of 119 ms and 124 ms, respectively (Fig. 1B).

### Song similarity quantification

The entire song-learning phase was monitored via audio recordings of song practice using SAP 2011[41]. At the end of the learning phase, similarity to the tutor song and syllable rate of the learned songs were analyzed. For calculating the similarity to the tutor song, we set SAP 2011 parameters to 'asymmetric' and 'time-courses' to capture if any syllables from the tutor song had been copied irrespective of their sequential order. Song similarity was then determined as the naturally occurring similarity across repetitions of the bird's own song when ten different repetitions were compared (see ref. 34).

Syllable rate was determined by using sound-envelope-based amplitude thresholding to segment syllables (200-250 syllables per bird recorded at 91–117 dph, $n = 33$ birds). After segmentation, the number of syllables detected was added up and divided by the sum of syllable durations of all syllables detected:

$$\text{Syllable rate} = \frac{\text{number of syllables}}{\text{sum(syllable duration)}} \quad (1)$$

### Bird's own song and female call recordings used in electrophysiological experiments

Songs of male birds were recorded with SAP 2011 at 44.1 kHz. Male birds were placed individually in sound-attenuated chambers, accompanied by a female bird. After a brief habituation period, male juveniles produced songs that were then used as BOS playbacks for intracellular recordings. The female call was extracted from a vocal interaction period between an adult female and a juvenile male bird. The same female call was used for all intracellular recordings. For the experiments with adult birds, female calls were recorded during natural interaction between the male and the female. In total, a stack call from two different females was used as playbacks. Each adult male bird was only presented with the call playback from a female it was housed with. The sound recordings were normalized, band pass filtered (300–14,000 Hz), and a 20 kHz pure tone (outside zebra finch hearing range)[62] was embedded for the duration of the call to detect call onsets and offsets.

### Tracking female vocalizations during song learning

Juvenile birds ($n = 6$ birds) were paired with an adult female ($n = 4$ birds) in a song-learning setup as described above. During the song-learning phase, juvenile birds were tutored with a song playback of the same adult bird. Female zebra finches were equipped with a microphone attached to their back with soft elastic leg straps[45]. The box microphone recording all sounds and the backpack microphone recording only the female sounds were then synchronized using a custom Matlab program. Recordings were made with the same Matlab program at 40 kHz sampling rate.

### Analysis of female vocalizations during song practice

We trained a neural network[46] to detect juvenile songs throughout the song-learning phase and a separate neural network was trained to recognize female vocalizations. We used the default setting in the Deep Audio Segmenter (DAS) and used 4 temporal convolutional network (TCN) blocks for training. As a quality measurement, we ensured that the precision of the network was at least 80%, and recall was set to at least 68%. If the precision score of the neural network per bird was below 80%, we added additional training data to reach the target.

We used one neural network for each bird to detect syllables from the box microphone recordings and female calls from the backpack microphone recordings every 2–3 days of the learning phase. Once the juvenile male song syllables and adult female calls were detected, we analyzed how many female calls occurred during song practice.

Song was considered to be at least 400 ms long event of consecutive syllables. To detect how many calls occurred during song practice, a time window of interest was specified as onset the onset of the first syllable (excluding introductory notes), and the offset of the last syllable of the song. To detect separate songs the end of each song was defined by at least 350 ms of silence between syllables.

A call was considered to be song practice-related if the onset of the call was detected within this window.

To evaluate, if female birds call during song practice, a call ratio was calculated to measure the number of calls throughout the duration of the song:

$$\text{Call ratio(nr/sec)}$$
$$= \frac{\text{Number of call onsets detected within the window of interest}}{\text{Duration of the window of interest (sec)}}$$
$$(2)$$

### Analysis of syllable similarity to tutor song after female call perturbation

We used the dataset of monitored female vocalizations during the song-learning phase (Fig. 3) and detected which juvenile songs

received female calls during song every 7–12 days. We then extracted 300 ms long snippets of the song directly after the female call. These snippets included the next syllable after the female call. For control, we also extracted 300 ms long snippets of unperturbed songs with the same onset and offset timing during the song as the perturbed snippets. We analyzed 264 song snippets of four birds, bird 5 was not analyzed due to a lack of female calls during the song. Similarity analysis was done as described above.

## Surgery

For head-fixed awake electrophysiological recordings, male zebra finches ($n = 6$ birds) were implanted with a head plate for fixation. First, male birds were anesthetized with isoflurane (concentration: 1–3% isoflurane, 97–99% oxygen). After careful incision, the skull was exposed and the trabecular bone structure above the area of interest was removed with a dental drill (carbide bur, FG ¼, Johnson-Promident). Second, the robust nucleus of the arcopallium (RA) was targeted according to coordinates (head angle: 65°, bifurcation of the mid-sagittal sinus was used as a reference point; RA: posterior 1.85 mm, lateral 2.25 mm, ventral 1.8 mm; HVC: anterior 0.2 mm, lateral 2.3 mm, ventral 0.2 mm, for extracellular recordings head angle was 45°, HVC coordinates: 0.3 mm anterior, 2.35 mm lateral). Third, a carbon fiber electrode (Kation Scientific, LLC) was lowered and RA was identified based on its characteristic firing pattern[63,64]. The location of HVC was then confirmed via antidromic stimulation from RA[38]. Fourth, the RA stimulating electrode and a head plate were implanted using light acrylic and dental cement (Paladur, Heraeus). For signal reference, a small craniotomy above the cerebellum was made and a ground wire (0.05 mm bare, silver) was placed between the skull and the dura mater covering the cerebellum. Lastly, all craniotomies were sealed with a silicone elastomer (Kwik-Cast) to prevent desiccation.

## Electrophysiological recordings

Intracellular recordings were performed after one recovery day after surgery. Animals were placed in a small foam-lined container to restrict movement. The head of the animal was fixed to allow stable access to the craniotomy.

## Intracellular recordings in awake head-fixed juvenile birds

Sharp intracellular electrodes (borosilicate glass with filament, 0.1 mm diameter) were pulled using a micropipette puller (Model P-97, Sutter Instrument) and backfilled with potassium acetate (concentration: 3 M). The identified location of HVC was secured by building a well with silicone elastomer around the craniotomy, which was then filled with phosphate-buffered saline (PBS). Dura was carefully removed using a dura-pick to gain access to HVC. Intracellular recordings were performed by lowering the glass pipette into HVC with a micromanipulator (model MP-285A, Sutter Instrument), no more than 10 micrometers at a time. The current depth of the electrode was assessed by the micromanipulator. Every time the electrode was lowered, a brief buzzing pulse (10–20 ms) was elicited to enter the cell membrane. Cells selected for further analysis had at least 30 mV action potentials with a resting membrane potential below −50 mV and the recording lasted for at least 3 min. To identify the neuron type, antidromic stimulation was performed and only cells with low jitter antidromic spikes were considered to be HVC$_{RA}$ neurons. Based on the low firing rate[39] of the recorded neurons and the characteristic spike waveform we concluded that none of the cells recorded were inhibitory interneurons.

## Intracellular recordings in freely moving juvenile birds

We used juvenile birds that were normally reared within our breeding colony up to 60 days post-hatch. After 60 days post-hatch, birds were placed into an adjacent aviary where several adult female birds were already housed. All juvenile birds maintained visual and auditory contact with their biological parents. To record intracellularly from a freely moving bird, we assembled and implanted a motorized microdrive[39]. After the surgery (as described above), we allowed the juvenile bird to recover for the next 2–3 days until the bird was singing with the implanted microdrive. Intracellular recordings were performed as described above. Upon successful recording of a cell, the juvenile bird was presented with one of the adult female birds from the aviary to motivate singing behavior. The first 2–3 motifs of directed singing were left unperturbed. For subsequent motifs, we triggered the playback of a female call manually to sample the neural activity of female call perturbations during song production.

## Extracellular recordings in awake head-fixed adult birds

Neuropixels probes[65,66] were used to record in awake, head-fixed adult birds. Only recording sites that covered the ventral length of HVC were considered. A New Scale micromanipulator was used to insert a single probe at an angle of 10°. Once it reached the desired depth, it was stabilized for 20 min. The recording was carried out using an external reference and shared ground-reference configuration. Then, playbacks of a female call were presented every 1.5 s for a minimum of 60 trials. Audio and neural signals were both acquired simultaneously and aligned offline.

## Data analysis

All statistical tests used are specified in the main text. If the statistical test for normally distributed data were applied, the dataset was first confirmed to be normally distributed by an Anderson–Darling test. All medians, means ± standard deviations are reported in the text.

**Firing rate.** To determine whether a cell responded in relation to a playback of a female call, we analyzed the change in firing rate, during call playback (call duration +50 ms[67]) and compared it to the firing rate of a silent period of the same duration that occurred before the call (offset of silent period was aligned to call onset −50 ms to avoid any overlap). The firing rate was calculated for every trial as a number of spikes per given time window. For every cell, the average firing rate was calculated across all trials of the same auditory stimulus or silent period, respectively. For intracellular recordings, spikes were detected at a 15 mV threshold above the baseline membrane potential. For extracellular recordings, data was acquired using SpikeGLX, spikes were detected and single units were sorted using Kilosort 2.5. Doubled counted spikes were removed using ecephys spike sorting repository. Sorted clusters were manually curated using phy, and single units were obtained after calculating quality metrics considering an interspike interval violation index (isi) >0.25, amplitude cutoff distribution <0.3, and presence ratio >90% [68]. To differentiate projection neurons from interneurons, we used previously described spontaneous firing rates typical for both types of neurons. We considered neurons with a spontaneous firing rate <=2 Hz as HVC projection cells, and correspondingly, cells with a firing rate >2 Hz as HVC interneurons.

**Spiking precision.** To assess how time-locked the cell is firing, a precision score was calculated as previously described in ref. 34: Differences in latencies between all spikes, across all trials of the same auditory stimulus were calculated and the mean latency difference was extracted. To test for significance, a permutation test was used: Spikes were shuffled across trials 1000 times preserving the statistical distribution of their occurrence and then the mean latency from the shuffled data was extracted. Responsive cells were defined as having a precision score outside of 95% of the shuffled mean latency distribution.

**Subthreshold correlation.** To specifically determine changes in subthreshold activity in relation to playbacks, we cut off the spikes detected at the 15 mV threshold above baseline membrane potential

and linearly interpolated the resulting membrane potential traces. The subthreshold activity during playback and the aforementioned silent period were compared. To analyze the stereotypy of the subthreshold activity during auditory stimuli, the correlation of subthreshold traces at a 0 lag across all trials of the same stimulus was calculated. An average value per cell across all trials of the same stimulus was calculated. The confidence interval for a population analysis was defined as ±1.96 standard error of the mean.

**Responsiveness to female call perturbation.** To determine whether a cell was responsive to a female call perturbation, we first detected the number of spikes during the female call perturbation +50 ms to allow for delayed auditory responses[67]. We then compared the number of spikes during the female call perturbation to the average number of spikes during the same epoch during the song without the female call. For control, we calculated the difference in a number of spikes during BOS renditions only. We then calculated a confidence interval of 1.96 standard deviations from the mean of the delta number of spikes during BOS. We marked any delta number of spikes during the female call perturbation In Fig. 6F if it exceeded the confidence interval boundaries.

**Statistical analysis**
- Neuronal data: For metrics that could be measured in individual trials (firing rate), we analyzed the data for each trial, controlling for neuron identity in order to account for any variability in the number of trials for each neuron. In metrics that measured properties across trials (precision), we analyzed the summary parameters for each neuron, controlling for bird identity. In order to account for the nested structure (neurons within birds) in continuous data, we fitted a linear mixed model by maximum likelihood estimation (changes in neuron firing rate/spiking precision depending on condition (baseline or female call)). We used the following model specifications:

neural activity ~ 1 + condition + (1 | bird ID) + (1 | bird ID:cell ID)
- Behavioral data: To test for song similarity/similarity in syllable rate differences and account for the genetic background or early sensory experience of individual birds, we fitted a linear mixed model with the following model specifications:

Similarity (song or syllable rate) ~ 1 + condition + (1 | father ID) + (1 | father ID:bird ID).

Detailed results for individual statistical analysis can be found in Supplementary Table 1.

**Reporting summary**
Further information on research design is available in the Nature Portfolio Reporting Summary linked to this article.

## Data availability
Example data are available here https://github.com/vallentinlab/Female_songlearning. Due to space limitations of the public repository, the complete song and neural recordings will be made available upon request. Please contact the corresponding author. Source data are provided with this paper.

## Code availability
All custom scripts including example data are available here https://github.com/vallentinlab/Female_songlearning.

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

## Acknowledgements
We thank Fabian Heim, Tamir Eliav, Giacomo Costalunga, Jonathan Benichov and Aditi Agarwal for useful comments on a previous version of the manuscript; Ofer Tchernichovski for providing data from seven untutored birds; Lisa Trost and Petra Müschenborn for technical support; Constance Scharff for advice on the project and constant support, Jeanette Meinecke and Barbara Buhlmann for animal care and

husbandry, Philipp Norton for helping to acquire the backpack microphone data. This work was supported by the HORIZON EUROPE European Research Council (ERC)-2017-StG-757459 MIDNIGHT, the Deutsche Forschungsgemeinschaft VA742/2-1, and the Deutsche Forschungsgemeinschaft 327654276–SFB 1315—awarded to D.V.

## Author contributions

L.B. and D.V. conceived the study and designed the experiments, L.B. and C.M.G.G. conducted the experiments, L.B., Y.X., C.M.G.G., and D.V. analyzed the data, L.B. and D.V. wrote the manuscript, and D.V. acquired funding and supervised the project.

## Funding

## Competing interests

The authors declare no competing interests.
