## [Peer Review File · Nature Communications]

Female calls promote song learning in male juvenile zebra finchesREVIEWER COMMENTS

Reviewer #1 (Remarks to the Author):

Summary:

In this manuscript, the authors investigate the neural underpinnings of social influences on vocal learning. They document that the presence of an adult female enhances the acquisition of a tutor song, that female calling during song practice changes across song development, and that HVC projection neurons are responsive to female calls in juvenile (and adult) birds. The manuscript is well-written and the experiments address an important question (since social influences on vocal development are prominent across many species, including humans). However, there are a number of important concerns about the specific contribution of females to song development and about the robustness of the data that need to be addressed.

Major concerns:

Some of the data are preliminary in nature. In particular, sample sizes are small for the analysis of female calls during song development (n=2 females across 3-4 developing males) and of the responses of projection neurons to female calls in quiet (n=4 neurons in n=__ birds) and in singing juveniles (n=5 neurons in n=2 birds). I appreciate that these experiments are difficult to conduct, but more data are required to ensure the robustness of the results for this journal.

The authors tend to equate female calling with female “feedback” in many places of the manuscript without fully demonstrating that female calls can serve as feedback signals for vocal performance to shape vocal development. The critical experiment of experimentally playing back female calls contingently or non-contingently during vocal practice was not conducted. (Another experiment could be to house juvenile males with mute or deaf females, but the playback experiment is stronger.) This experiment is essential in demonstrating that female calls act as female signals.

-NOTE: a paper that might be of interest and could help the authors is Adret (2004), *Journal of Ethology*.

The authors should analyze the degree to which practice songs with vs. without female calls differ in their similarity to the tutor song or in stereotypy. This would further support the possibility that female calls serve as feedback signals. Changes in female calling over a juvenile male’s development are interesting but could be due to other factors (e.g., duration of housing), since no control group was conducted for this experiment.

More information about how juveniles were raised before song tutoring is required because it has implications for the interpretation of the results. Based on the Methods section, it sounds like juveniles were housed with both parents before they were isolated for song tutoring (<35 dph). It is evident that a substantial amount of song can be learned before 30 dph (e.g., Roper and Zann, 2006). Therefore, it will be important for the authors to analyze a few things with regard to the father's song. For example, because it is not clear if siblings from the same nest were divided across conditions, the authors need to ensure that the fathers of the birds that were housed with females during development did not produce songs that were MORE similar to the tutor song than the fathers of the birds that were not housed with females during development. In other words, the results reported in the manuscript might not be due to differential learning of the tutor song but due to differential similarities of the juveniles' father's songs to the tutor stimulus. In addition, it will also be important for the authors to analyze the similarity of the adult pupil's songs to their father's songs.

The similarity scores (i.e., to the tutor stimulus) for both sets of tutored birds is quite low (though there is quite a bit of variation). To ensure that learning is occurring, the authors should provide a control analysis. For example, a common and useful control measurement is the degree to which the songs of untutored birds resemble the tutor stimulus.

Because females were housed with juveniles throughout development, the experiments do not rule out the possibility that females are enhancing the sensory acquisition of the tutor song. At minimum, the authors should include a discussion of this alternate possibility. Better than this would be that the authors conduct experiments (see above) to more specifically test the hypothesis that female calls serve as feedback signals for juvenile song performance.

Minor comments:

Lines 122-141: I suggest that the authors first describe the characteristics of the tutor song and then describe differences between juvenile birds raised with vs. without a female. That way readers would understand the reason for these analyses sooner in the text.

Figure 1: Can the females trigger song playback? (it seems this way back on the images and description) If so, is experimental variation in song imitation due to variation in the degree to which the juvenile males (vs. females) activated song playback?

Figure 1: The gaps in the tutor stimulus are atypically long for a zebra finch motif. It is interesting to point out here that the gaps in the tutored birds' adult song are more species-typical, suggesting that they modify their songs in species-typical ways (which is consistent with various published

studies). (Or it could be that their gaps are more similar to their father's song than to the tutor stimulus.)

Line 169-171: It is important to assess female calling for this fourth juvenile male, because this would lend further insight into the role of female calling in vocal development. Sample sizes are already really small for this analysis.

Line 174-176: Did the amount of female calling increase over development because the amount of vocal practice by juveniles increased over development?

Line 185-187: It is interestingly that the call ratio for the female housed with bird 3 was high from the beginning. This finding further supports the importance of assessing the degree to which female calling was differentially associated with the "quality" of juvenile song under these various conditions. Maybe this female did not differentially call in response to the quality of bird 3's vocalization...

Line 210-213: 4 neurons from how many birds?

Line 238-241: The authors should not completely dismiss this statistical trend ($p=0.06$). The data in Figure 3H is quite compelling.

Line 281: the citation should be 39?

Line 339-343: these sentences seem out of place, I suggest deleting.

Reviewer #2 (Remarks to the Author):

The manuscript by Bistere et al. is a beautiful study of how social context can impact song learning. Feedback has been shown to be very important in learning language in humans and it is exciting to see this replicated in birds, where the neural mechanisms can be studied. The authors performed bold and difficult experiments that provided sufficient evidence for a role of female calls in shaping the song learning trajectory of juvenile male birds. The comments I enumerate below can be addressed by relatively minor changes to the manuscript, including additional discussion.

Line 58: there seems to be a missing word ('The'?) at the beginning of the second sentence.

For Figure 2, why did the authors decide to use the same female for bird 1 and bird 3? If the experiment was done first on 'bird 1', the female would be older for the experiment done on 'bird 3' – would the age of the female matter in what kind of feedback they give, or in how the juveniles receive this feedback?

Also in Figure 2 – for ;bird 3', the female vocalizations start at a higher rate. It is true that they do not increase over time, but there is frequent feedback from the start. I wonder if the female feedback could serve as a signal that the song is good, and the juveniles stop refining it when they get frequent feedback?

Also for Figure 2 and related text, I would suggest using 'juvenile 1, 2,3' instead of 'bird 1,2,3'. This would improve clarity (the females are also birds).

Line 242: 'As' instead of 'as'?

Lines 252-254: I am confused about the conclusion presented here. Is this based on 8 cells receiving more inhibition vs 7 cells receiving more excitation?

Figure 3 – I am not sure what the experiment with BOS and female call adds. Hearing BOS without producing it must be confusing for the juvenile, and does not really reproduce the conditions during singing. Experiments in Figure 5 are much better at illustrating the interaction between female calls and juvenile singing.

In the song learning experiments, did the authors verify that all birds peck for tutor song at least 40 times? When the female is present, does she ever peck for the song?

Are female vocalizations sufficient to improve song learning? If in Figure 1 the authors replace the female bird with female calls that increase in frequency over days, do they expect the juvenile to learn better than if no female calls are played back? I am NOT asking for this experiment to be done, but maybe the authors could speculate in the discussion?

The authors should check the punctuation in the manuscript; some commas and periods are missing.

Reviewer #3 (Remarks to the Author):

This is an interesting manuscript providing novel and exciting evidence for the influence of social feedback on vocal learning in songbirds. In the present study, Bistere and colleagues investigated the impact of social feedback from female zebra finches on the song learning performance of juvenile zebra finches. They found that the presence of adult female zebra finches led to more accurate imitation of the tutor song in juvenile zebra finches. As juvenile males improved their song performance, females emitted more calls, indicating that they are able to directly influence the progress of juveniles' song learning through their feedback. Next, the authors performed intracellular recordings of HVC projection neurons in listening and singing zebra finches. They found that female vocalizations modulate neural activity in HVC during passively listening and singing in juvenile birds, suggesting a social feedback mechanism for guiding developmental song learning.

Remarkably, this is the first study providing evidence that female vocal feedback significantly contributes to developmental song learning in a songbird, and that vocal input other than the tutor song can influence the neural circuits involved in song learning and production. Consequently, this study will lead to a range of new research questions on the role of social feedback on song learning and the underlying neural network. The paper is thus of general interest and will clearly appeal to the large and diverse readership of Nature Communications

Overall, this is an impressive, highly original, and hypothesis-driven study! The methodological approach appears sound and solid, and the number of animals is sufficient. The data are convincing, well-analyzed, and clearly and concisely presented. The manuscript is well-written and was a pleasure to read. I believe this manuscript is actually almost "ready to go." I have just two minor comments that the authors might want to address.

1. Figure 1D nicely shows a significantly higher song similarity for juvenile birds raised with females. However, the overall mean song similarity appears to be low in both cases. The authors might want to discuss this finding also in terms of what level of song similarity would be expected in birds raised with their father?

2. In terms of design, Figure 4 stands out from the other, much more detailed Figures, which have many subplots. This is particularly striking because the information conveyed by Figure 4 does not appear to be as important as that conveyed in the other Figures. The authors might want to consider combining Figure 4 and 5.

Point-by-point response

We would like to thank the reviewers for their insightful comments concerning our manuscript. Based on the reviewers' concerns we conducted additional experiments and analyses to further strengthen our findings.

REVIEWER COMMENTS

Reviewer #1 (Remarks to the Author):

Summary:

In this manuscript, the authors investigate the neural underpinnings of social influences on vocal learning. They document that the presence of an adult female enhances the acquisition of a tutor song, that female calling during song practice changes across song development, and that HVC projection neurons are responsive to female calls in juvenile (and adult) birds. The manuscript is well-written and the experiments address an important question (since social influences on vocal development are prominent across many species, including humans).

We thank the Reviewer #1 for the overall positive assessment of our work.

However, there are a number of important concerns about the specific contribution of females to song development and about the robustness of the data that need to be addressed.

Major concerns:

Some of the data are preliminary in nature. In particular, sample sizes are small for the analysis of female calls during song development (n=2 females across 3-4 developing males) and of the responses of projection neurons to female calls in quiet (n=4 neurons in n=__ birds) and in singing juveniles (n=5 neurons in n=2 birds). I appreciate that these experiments are difficult to conduct, but more data are required to ensure the robustness of the results for this journal.

We thank the reviewer for pointing out this valid concern. We undertook a major effort to replicate these challenging experiments and now included a larger sample size for most data sets. All previous reported results remained valid and we can draw even stronger conclusions due to more statistical power. In detail,

- *analysis of female calls during song development*: We raised two additional juveniles with two additional females which resulted in n = 4 females across 6 developing males. The sample size was changed in the text (page 10 line 192-193) and the revised results are now highlighted in Figure 3 C-F (juvenile 3 and 5) and updated within the main text (page 10 line 211-219, 221-231 and 233-243). In line with the previously reported results, juveniles receiving female calls that were correlated with the time of song practice produced a song

that was a closer copy to the tutor song compared to juveniles raised with non-correlated female calls. To further assess the quality of song performance we calculated the mean similarity of songs from birds raised in isolation to our tutor song playback and determined this value to be the threshold for learners versus non-learners. Using this criterion, we found that bird 4 and 5 did not learn the tutor song. This new analysis is now displayed in Figure 3D and the results are integrated on page 10 line 228-231.

- analysis of the responses of projection neurons to female calls in quiet and in singing juveniles (n=4 neurons in n=__ birds):

We now added the number of birds in which we intracellularly recorded HVC projection cells in response to female calls (4 identified RA projecting neurons and 11 unidentified HVC projection cells) of 6 juvenile birds during listening (page 13 line 259-262). We apologize for this confusion.

Additionally, we performed further intracellular recordings of HVC projection neurons in singing juvenile birds. We are now reporting results from 9 HVC projection neurons of 4 birds (Figure 6C-F). The sample size increase allowed us to conduct an additional statistical assessment of the data and we found that in 6 out of 22 female call perturbations the neural activity significantly changed compared to regularly occurring activity when the bird is singing without female call interruption (Figure 6F). We now also report this finding in the text (page 18 line 355-359).

The authors tend to equate female calling with female “feedback” in many places of the manuscript without fully demonstrating that female calls can serve as feedback signals for vocal performance to shape vocal development. The critical experiment of experimentally playing back female calls contingently or non-contingently during vocal practice was not conducted. (Another experiment could be to house juvenile males with mute or deaf females, but the playback experiment is stronger.) This experiment is essential in demonstrating that female calls act as female signals.
-NOTE: a paper that might be of interest and could help the authors is Adret (2004), Journal of Ethology.

We agree with Reviewer #1 that equating female calls with female feedback is not always suitable. We therefore rephrased parts of the manuscript where we think ‘feedback’ was used too ambitiously (page 3, lines 79, 81, 91, page 4 line 97, page 9 line 181). Due to the newly conducted experiments which strengthened the result that song-correlated female calls lead to improved tutor song imitation, we think that to an extent the use of the term ‘feedback’ is justified hereafter. We agree with Reviewer #1 that to conduct a song learning experiment with contingent and non-contingent female call playbacks would ultimately resolve whether it is feedback targeted to specific parts of song practice. To further underline that female vocalizations are key to song learning we appreciate the suggestion of Reviewer #1 to refer to and discuss the work by Adret 2004 (page 9 line 186-187, page 20 line 392-400). This work highlights that during social interactions between songbirds’ vocal exchanges can be sufficient and independent of visual inputs and thus have the potential to serve as feedback to guide song learning.

The authors should analyze the degree to which practice songs with vs. without female calls differ in their similarity to the tutor song or in stereotypy. This would further support the possibility that female calls serve as feedback signals. Changes in female calling over a juvenile male's development are interesting but could be due to other factors (e.g., duration of housing), since no control group was conducted for this experiment.

We thank Reviewer #1 for the thoughtful suggestion of comparing the song similarity with or without female calls. We have now included an additional figure (Supplementary Figure 4) and refer to it in the main text (page 18 line 362-365) to address this issue. One hypothesis is that the song similarity change occurs within the ongoing syllable immediately during female call interruption. Acoustically the song and the female call cannot be separated and we are not able to perform this analysis. Alternatively, song similarity might change in response to the female call with a delay. To test this, we detected which juvenile songs received female feedback and extracted 300 ms long snippets of song directly after the female call. For control, we also extracted 300 ms long snippets of unperturbed songs. We analyzed 264 song snippets of four birds that were raised with a female wearing a microphone backpack. We did not find significant differences in song similarity to tutor song across all days between song snippets after call and song snippets of unperturbed songs (Supplementary Figure 4). We then compared the similarity scores of perturbed and unperturbed song snippets each day and only found a difference in Bird 3 at 64 days post hatch. The possibility remains that the impact of the female calls on song learning may not be immediate but rather serve as persistent feedback that is integrated over several repetitions of practiced song (as suggested by the song performance outcome that is correlated with a female call increase (Figure 3)). We added this speculation in the text (page 18 line 365-368).

More information about how juveniles were raised before song tutoring is required because it has implications for the interpretation of the results. Based on the Methods section, it sounds like juveniles were housed with both parents before they were isolated for song tutoring (<35 dph). It is evident that a substantial amount of song can be learned before 30 dph (e.g., Roper and Zann, 2006). Therefore, it will be important for the authors to analyze a few things with regard to the father's song. For example, because it is not clear if siblings from the same nest were divided across conditions, the authors need to ensure that the fathers of the birds that were housed with females during development did not produce songs that were MORE similar to the tutor song than the fathers of the birds that were not housed with females during development. In other words, the results reported in the manuscript might not be due to differential learning of the tutor song but due to differential similarities of the juveniles' father's songs to the tutor stimulus. In addition, it will also be important for the authors to analyze the similarity of the adult pupil's songs to their father's songs.

We agree with Reviewer #1, that early life experience might have influenced how well the juvenile birds copy the tutor song. We therefore describe the early life experience in more detail in the method section and refer to Roper & Zann 2006 when describing vocal development in zebra finches in the discussion (page 20 line 396-397). Further, we added

more details about the housing and upbringing conditions in the method section (page 23 line 484-490). We also added the comparison between learnt song after tutoring and the genetic fathers' song of all birds for which we had access to their genetic father. We found that the song of juveniles raised without a female was more similar to their father's song whereas the song of juveniles raised with a female was equally similar to the tutor and the father song (new Figure 2). We also compared the similarity of fathers' songs to tutor song playback and did not find a difference in the similarity between fathers from birds of both conditions (fathers from birds raised with female: song similarity mean=30.29±8.33, median=28.89; fathers from birds raised alone: song similarity mean=21.28±6.61, median=20.19; Wilcoxon rank sum test, p-value=0.17, Figure R1). These findings suggest that the presence of a female can overcome the genetic predisposition of the fathers' song, in line with what has been reported in Mets et al. 2017 for the presence of a male tutor. These new insights are now integrated in the manuscript (page 7 line 149-167) and illustrated in a new Figure 2.

Figure R1: Similarity of fathers' song to tutor song playback.

The similarity scores (i.e., to the tutor stimulus) for both sets of tutored birds is quite low (though there is quite a bit of variation). To ensure that learning is occurring, the authors should provide a control analysis. For example, a common and useful control measurement is the degree to which the songs of untutored birds resemble the tutor stimulus.

We thank Reviewer #1 for this suggestion. We have now compared the similarity score of our tutored birds to untutored birds. We are grateful to have received example recordings of untutored birds from Ofer Tchernichovski. These birds were raised in isolation and had a similarity score to our tutor song of 26.5 %. We defined this as the threshold for learning (see Figure 3 D) and separated birds into learners versus non-learners (page 10 line 228-231).

Because females were housed with juveniles throughout development, the experiments do not rule out the possibility that females are enhancing the sensory acquisition of the tutor song. At minimum, the authors should include a discussion of this alternate possibility. Better than this would be that the authors conduct experiments (see above) to

more specifically test the hypothesis that female calls serve as feedback signals for juvenile song performance.

We agree with the Reviewer #1, that females could enhance the sensory acquisition of tutor song. We now include this possibility in the discussion (page 20 lines 401-408, page 21 line 423-426).

Minor comments:

Lines 122-141: I suggest that the authors first describe the characteristics of the tutor song and then describe differences between juvenile birds raised with vs. without a female. That way readers would understand the reason for these analyses sooner in the text.

We have now added information about the specifics of the tutor song playback on page 23 line 499-504 in the Method section.

Figure 1: Can the females trigger song playback? (it seems this way back on the images and description) If so, is experimental variation in song imitation due to variation in the degree to which the juvenile males (vs. females) activated song playback?

We agree with Reviewer #1, that female birds could have also triggered song playbacks. However, our experimental design only permitted 40 playbacks, that were triggered by key pecks per day. We carefully observed, that all birds (birds with female and also birds raised alone) fulfilled the 40-key peck-per-day quota. Therefore, if females pecked the keys too, this would lead to less key pecks for the juvenile that they were housed with. The results of our study however show, that birds raised in female presence had a higher song similarity to tutor song. We therefore assume, that even if females pecked the keys for playback, this variability had a negligible effect on our results.

Figure 1: The gaps in the tutor stimulus are atypically long for a zebra finch motif. It is interesting to point out here that the gaps in the tutored birds' adult song are more species-typical, suggesting that they modify their songs in species-typical ways (which is consistent with various published studies). (Or it could be that their gaps are more similar to their father's song than to the tutor stimulus.)

We agree with the Reviewer #1, that the gap duration result from our experiment is interesting and we have added more information about how it aligns with previous studies showing that zebra finches have a species-specific bias towards the gap duration, that is independent of the gap duration of their tutor (Araki et al. 2016, *Science*, Logan et al. 2023, *Developmental Science*) (page 5 line 136-138).

Line 169-171: It is important to assess female calling for this fourth juvenile male, because this would lend further insight into the role of female calling in vocal development. Sample sizes are already really small for this analysis.

We agree with Reviewer 1#, that the sample sizes for this experiment were initially small. We have now added 2 more birds (see Figure 3) to yield robust results. As for bird 4 (it has

now been renamed bird 6), it is unfortunately not possible to detect song in an automated way, since the song only consists of harmonic stacks that were too similar to calls (see Supplementary Figure 2).

Line 174-176: Did the amount of female calling increase over development because the amount of vocal practice by juveniles increased over development?

To address this concern, we quantified the number of songs per day for each bird (illustrated in Figure 3) and did not observe an increase in singing rate per day. This result is now added to the main text (page 10 line 217-219) and the Supplementary Figure 2.

Line 185-187: It is interestingly that the call ratio for the female housed with bird 3 was high from the beginning. This finding further supports the importance of assessing the degree to which female calling was differentially associated with the "quality" of juvenile song under these various conditions. Maybe this female did not differentially call in response to the quality of bird 3's vocalization...

We agree with Reviewer #1, that female birds might call in relation to juvenile songs differently depending on specific song features. However, we consider the assessment of the spectral features that evoke more calling to be outside of the scope of this study. Additionally, we now analyzed the similarity of syllables following female call interruptions. We could not find a significant in- or decrease in song performance shortly after female calls during song practice (see response to Major concern 3).

Line 210-213: 4 neurons from how many birds?

We added the number of birds on page 13 line 259-262.

Line 238-241: The authors should not completely dismiss this statistical trend ($p=0.06$). The data in Figure 3H is quite compelling.

We agree with Reviewer #1 that the statistical trend that HVC projection neurons might increase their firing rate in response to BOS when accompanied with a female call. We mention this trend in the text but would not like to argue further. Given that we added more data in the song context condition it seems more interesting to us that we can now report significant changes in response to female calls during song practice.

Line 281: the citation should be 39?

The reference has been changed.

Line 339-343: these sentences seem out of place, I suggest deleting.

These lines have been deleted.

Reviewer #2 (Remarks to the Author):

The manuscript by Bistere et al. is a beautiful study of how social context can impact song learning. Feedback has been shown to be very important in learning language in humans and it is exciting to see this replicated in birds, where the neural mechanisms can be studied. The authors performed bold and difficult experiments that provided sufficient evidence for a role of female calls in shaping the song learning trajectory of juvenile male birds. The comments I enumerate below can be addressed by relatively minor changes to the manuscript, including additional discussion.

We thank Reviewer #2 for the positive feedback.

Line 58: there seems to be a missing word ('The'?) at the beginning of the second sentence.

We have added 'the' in line 58 and thank Reviewer #2 for pointing it out.

For Figure 2, why did the authors decide to use the same female for bird 1 and bird 3? If the experiment was done first on 'bird 1', the female would be older for the experiment done on 'bird 3' – would the age of the female matter in what kind of feedback they give, or in how the juveniles receive this feedback?

Initially, we wanted to control for the learning experience of each juvenile and pair them with the same female. Based on the suggestion of Reviewer #1 we now trained additional juveniles with additional females. Therefore, our sample size increased and although a larger set of females with diverse backgrounds and ages were used, results remained valid and even became more robust. See page 10 line 211-218 and Figure 3.

Also in Figure 2 – for 'bird 3', the female vocalizations start at a higher rate. It is true that they do not increase over time, but there is frequent feedback from the start. I wonder if the female feedback could serve as a signal that the song is good, and the juveniles stop refining it when they get frequent feedback?

We thank Reviewer #2 for this interesting observation. With access to a larger data set we can now conclude that a consistently high female call rate during song practice does not result in a juvenile song that is not refined any further (see Figure 3 panel C juvenile 4). To underline this statement, we also extracted songs from juvenile 4 (former bird 3) and juvenile 5 on four days during the critical song learning period to show that the juveniles undergoes song changes across development (Figure R2).

Figure R2: Song development of Juvenile 4 and Juvenile 5. Although juvenile 4 did not receive correlated female feedback, similarity of BOS to BOS of the last day of training was increasing during development. Juvenile 5 received very low female feedback and sang songs with great variability during development.

Also, for Figure 2 and related text, I would suggest using 'juvenile 1, 2,3' instead of 'bird 1,2,3'. This would improve clarity (the females are also birds).

We agree with Reviewer #2 and have changed "bird" to "juvenile".

Line 242: 'As' instead of 'as'?

Has been changed.

Lines 252-254: I am confused about the conclusion presented here. Is this based on 8 cells receiving more inhibition vs 7 cells receiving more excitation?

We apologize for the confusion. We now referenced the comparison correctly and hope the conclusions are stated more clearly (page 14 line 299-302).

Figure 3 – I am not sure what the experiment with BOS and female call adds. Hearing BOS without producing it must be confusing for the juvenile, and does not really reproduce the conditions during singing. Experiments in Figure 5 are much better at illustrating the interaction between female calls and juvenile singing.

We agree with Reviewer #2 that experiments in singing juveniles more directly answer the question of how female feedback changes neural activity during song production. Therefore, it is worth noting that our presented data in Figure 6 are the first intracellularly

recorded HVC projection neurons from singing juvenile zebra finches. Given the challenges to achieve these recordings, we wanted to provide a larger data set. Since songbird HVC neurons can have mirror-neuron like activity when listening to a birds' own song (Prather et al. 2008, *Nature*) that is very similar to the neural activity during song production, we decided to add the results of these experiments.

In the song learning experiments, did the authors verify that all birds peck for tutor song at least 40 times? When the female is present, does she ever peck for the song?

The key pecking quota per day was fulfilled by all birds (birds pecked the keys 40 times per day). The female birds were also able to peck the keys and we could not control how often keys were pecked by the female or the juvenile. However, previous studies have shown, that juvenile males learn to copy the tutor song better when they peck the keys compared to passive playbacks of tutor song only (Derégnaucourt et al. 2013, *Journal of Physiology -Paris*) In that scenario, we would expect the birds raised with females to acquire a worse copy of tutor song than the birds in alone condition if the female pecks the keys. Our results, however, show that juveniles raised with a female learn to copy the tutor song better, so we can assume the possibility of females pecking the keys to be negligible. See also response to Reviewer #1 (minor comment 2).

Are female vocalizations sufficient to improve song learning? If in Figure 1 the authors replace the female bird with female calls that increase in frequency over days, do they expect the juvenile to learn better than if no female calls are played back? I am NOT asking for this experiment to be done, but maybe the authors could speculate in the discussion?

We thank Reviewer #2 for this suggestion. We now added a short speculation in discussion (page 21 line 423-426) and take this as an inspiration for future experiments.

The authors should check the punctuation in the manuscript; some commas and periods are missing.

Checked.

Reviewer #3 (Remarks to the Author):

This is an interesting manuscript providing novel and exciting evidence for the influence of social feedback on vocal learning in songbirds. In the present study, Bistere and colleagues investigated the impact of social feedback from female zebra finches on the song learning performance of juvenile zebra finches. They found that the presence of adult female zebra finches led to more accurate imitation of the tutor song in juvenile zebra finches. As juvenile males improved their song performance, females emitted more calls, indicating that they are able to directly influence the progress of juveniles' song learning through their feedback. Next, the authors performed intracellular recordings of HVC projection neurons in listening and singing zebra finches. They found that female vocalizations modulate neural activity in HVC during passively listening and singing in juvenile birds, suggesting a social feedback mechanism for guiding developmental song learning.

Remarkably, this is the first study providing evidence that female vocal feedback significantly contributes to developmental song learning in a songbird, and that vocal input other than the tutor song can influence the neural circuits involved in song learning and production. Consequently, this study will lead to a range of new research questions on the role of social feedback on song learning and the underlying neural network. The paper is thus of general interest and will clearly appeal to the large and diverse readership of Nature Communications

Overall, this is an impressive, highly original, and hypothesis-driven study! The methodological approach appears sound and solid, and the number of animals is sufficient. The data are convincing, well-analyzed, and clearly and concisely presented. The manuscript is well-written and was a pleasure to read. I believe this manuscript is actually almost "ready to go." I have just two minor comments that the authors might want to address.

We thank Reviewer #3 for the positive feedback to our manuscript.

1. Figure 1D nicely shows a significantly higher song similarity for juvenile birds raised with females. However, the overall mean song similarity appears to be low in both cases. The authors might want to discuss this finding also in terms of what level of song similarity would be expected in birds raised with their father?

We thank Reviewer #3 for this suggestion. We agree that song similarities seem to be lower compared to other studies. Due to animal protocol regulations we raise juveniles in larger sound attenuated boxes with more enrichment compared to previous studies. This difference might result in different temporal dynamics of pecking the keys and thus change the overall dynamics of learning. However, we are only comparing song learning across animals raised in our setup. Based on the Reviewer #3 we now include a comparison of song similarity of songs from isolated birds to tutor song (see Figure 3 D) and use this as a threshold for juveniles that did not learn the tutor song (page 10 line 228-231).

2. In terms of design, Figure 4 stands out from the other, much more detailed Figures, which have many subplots. This is particularly striking because the information conveyed by Figure 4 does not appear to be as important as that conveyed in the other Figures. The authors might want to consider combining Figure 4 and 5.

We thank Reviewer #3 for this comment. Figure 4 (now Figure 5) has now been reformatted.

REVIEWER COMMENTS

Reviewer #1 (Remarks to the Author):

I commend the authors for their additions and improvements to the manuscript. The analyses of similarity to father song, the addition of call analyses of two more birds housed with females, and the collection of more data from HVC projection neurons in juveniles have strengthened the manuscript. I continue to think that the authors are addressing an important and exciting issue in vocal learning. However, there continue to be concerns about analyses and quality of data. For example, it is unclear why the authors use very different types of analyses for similar types of neurophysiological data for juvenile and adult birds. I realize that this is a substantive set of comments, but addressing these concerns will increase the rigor of their analyses, instill greater confidence in the quality of the data and interpretations, and ultimately strengthen their exciting paper.

MAJOR COMMENTS:

Analysis of neural responses to female calls during song production:

- After conducting experiments designed to reveal the importance of female calls to juvenile song plasticity, the authors aim to identify neural populations that are sensitive to female calls during song practice. Overall, the data on the effect of call playback during song production are interesting but not compelling. And the authors overinflate the results. For almost 70% (15 out of 22) of the data, call playback leads to no change in the number of spikes. Additionally, in Figure 6F, the authors plot the absolute value of changes in spiking rates; when analyzing any change, plotting the absolute value can overestimate the effect. For example, if half the neurons increased in spike rate but the other half decreased in spike rate, then one would say that there wasn't a significant change in spiking activity (similar to other analyses in this paper); but if you look at the absolute change, the effect is going to look large. The authors need to adopt a more rigorous approach for this analysis.
- Similarly, since the authors emphasize differences in neural responses to female calls in adults (Vallentin and Long, 2015) and juveniles (this paper), it is important to conduct the same types of analyses across age groups. For example, one wonders what the analyses of adult responses would look like if the adult data were analyzed in precisely the same manner as analyzed here; my guess is that if you plot the absolute difference between control and experimental conditions in adults, you will likely have distributions of data that are above zero. More realistically, the authors should analyze juvenile responses in as similar a manner as possible to the adult data.
- More details about data collection and analyses for this experiment are required in the Methods. Given the narrative of the paper, this is a very important experiment, and readers should be able to understand the details of the approaches. (In the response to reviewers file the authors write that

“in 6 out of 22 female call perturbations the neural activity significantly changed compared to regularly occurring activity when the bird is singing without female call interruption (Figure 6F)” but it is not clear how statistical significance is computed for each of the 6 “female call perturbations”.)

Analysis of neural responses to female calls in awake but quiet adult and juvenile finches:

It is not clear why the analyses of auditory responses to female calls in awake but quiet adult finches and juvenile finches are so different. These are the same types of data. In many respects, the analyses of HVC projection neurons in juveniles are more intuitive, whereas the analysis of HVC neurons in adults seems to be tailored to highlight neural responses. As far as I can tell, there is no reason why the adult data cannot be analyzed in the same manner as juvenile data, especially in the same paper.

Analysis of the contribution of female calls to song improvement:

I appreciate the addition of two birds in the section on female vocal feedback increasing over development; these birds certainly help support the previous data. I also understand how this analysis helps segue into the neurophysiological experiments, but I continue to think that this experiment is preliminary. For example, only three of the five juveniles benefited from the presence of the female, and the juvenile that experienced the largest increase in call ratio (juvenile 3) did not produce a song that imitated the tutor song well. In a high profiles journal like this, a strong statement about the impact of female calls during song practice on song learning (e.g., in the Abstract) that is based on $n=5$ will become amplified for people who do not read the details of the paper. While I believe that the best solution to strengthen this paper for this journal is to continue to increase the sample size (e.g., $n=8-10$ juveniles paired with females that increase call rates over time vs. $n=8-10$ juveniles paired with females that do not increase call rates over time), one possibility is to reword their abstract and discussion in a manner that highlights the preliminary nature of the analyses.

Statistical models:

This should have been noted in the original submission but I realize that there are a number of places where statistics are inappropriate. My largest concern deals with the analysis of neurophysiological data. There is an increasing realization that statistics that treat each neuron within an individual as an independent sample are flawed (e.g., Yu et al., *Neuron*, 2022; Reed and Kaas, *Neural Networks*, 2010; Koerner and Zhang, *Brain Sciences*, 2017) and that mixed effects models are more appropriate for this type of data. To this end, the authors should analyze their data with some random factor; for example, bird ID should be added as a random factor in their statistical models for neurophysiological data. Right now, there is pseudoreplication in their analyses. For example, there is pseudoreplication when treating all 133 neurons in 5 birds as independent (Figure 5) as well as when treating the response of the same HVC neuron to multiple call perturbations (Figure 6). In the analysis of HVC responses to female calls during song

production, it is possible that all six neurons that demonstrate changes in spike rate in response to call playback came from the same individual or even the same neuron (see comment above about the ambiguity of these analyses), and the current statistical approach does not account for this.

Here I list various comments about statistical analyses, be it major or minor:

- lines 150-168: a within-individual analysis (not a two-sample t-test) should be used because each tutored bird has a similarity to the tutor and to the father. (It is possible that they actually ran a paired analysis for song similarity because they run a paired analysis for syllable rate later in the paragraph.) Because different birds can share a father and because genetics (or experience with a particular father) could affect learning, a mixed effects model with father ID as a random factor would be appropriate to analyze these data (similarity and syllable rate).
- After reading the analyses in lines 156-160, one expects a comparison of similarities in syllable rates to the playback stimulus vs. to the father's song (e.g., pupil syllable rate compared to playback and father's syllable rates). This is a minor suggestion but I do I think this analysis would be a useful addition.
- Lines 144-147: indicating that they are analyzing the distribution of differences here (if different from zero) would be useful (but see larger comment about mixed models)
- Lines 152-154: Justify why only a subset of juveniles are included in this analysis here. It seems like 5 out of 33 birds (15%) were excluded from this analysis, and it would be useful to briefly know why in the Results section.
- Lines 235-237 and lines 243-244: why are there comparisons between individual birds?
- Line 300-303: The authors can explicitly compare the degrees of change using a paired analysis (mixed effects model) instead of just making a qualitative statement.

Song similarity:

Line 514-517. This calculation escaped me on my first read through. Is there a strong justification for this formula? The vast majority of studies using SAP to quantify song learning simply by analyzing the similarity of BOS to tutor song (i.e., one of the variables in the numerator). It is not clear why the SAP similarity score is multiplied with the accuracy score and then normalized for these analyses, so a justification is important here. Because I suspect that other readers will similarly interpret the data in their figures in the same manner as I originally did, the authors should use a different name than "song similarity" in the main text and figures so people do not try comparing the song similarity scores in this paper to song similarity scores in other papers. In addition, given the prevalence of analyses of SAP similarity scores, it would be very useful to analyze and plot this data in a supplementary figure (or in the main text).

MINOR COMMENTS:

Abstract: line 40: I suggest replacing “as young birds” with “the males that they were housed with” because the “young birds” in the current wording could refer to the females themselves.

Abstract: line 40: replace “indicating” with “suggesting” since the former is too strong.

Line 44-49: “song-related” could refer to listening to song, so "singing-related" is better. Please replace.

Line 80: why limit this to “their offspring”? Are you specifically talking about mothers (that is not evident here)? “their offspring” can be replaced with "males" or "developing males"

Lines 117-119: can the authors say that the tutor song was acoustically distinct from the father’s song? This would be useful to say (depending on the degree to which this is true) for experiments on juveniles that were previously tutored by their fathers.

Line 126: Suggest adding a paragraph break here to make the results in lines 121-126 more impactful. Right now this paragraph is very long, and important results can get buried.

Lines 144-148: it seems more impactful to put these results BEFORE the results on syllable and gap durations. After reading the results on syllable durations, I wondered why these results were presented before the results on syllable rate (which are more compelling for learning).

Figure 1A legend: add sample sizes for each group in figure legend. This would generally be useful in legends throughout the manuscript.

Line 150-154: I wouldn't call these genetic predispositions, since they are based on experiences during development. Moreover, the authors should explicitly mention in the Results section that the juveniles were raised with fathers for some period of time and could have learned their father's song to some degree.

Figure 2A: It is hard to see which tutored bird is tutored with or without females. Maybe the authors could have orange vs. green backgrounds behind the panels (or a vertical orange or green line to the left of the spectrograms) to make the difference between birds more conspicuous.

Figure 2B: add label for which birds were tutored with or without females above the figure to make the figure easier to understand.

Lines 183-184: same comment about genetic predispositions rephrased as experience-dependent learning.

Lines 202-207: in parentheses, just list range without listing numbers per bird. This is one of the many places where there is too much text in parentheses.

Line 211: replace "learning" with "performance". The former requires that they analyze similarity to tutor song throughout development (which isn't a bad idea, but probably quite labor intensive).

line 214-217: awkward sentence, unclear how juvenile 5 fits into this or why other juveniles that were raised with different females were not compared here. For the reader to understand the full extent of these comparisons, the authors should first indicate which juveniles were housed with the same females. Or just simplify this sentence.

Line 220-222: I suggest rewording these sentences (or at least the first of these two). While I understand why the authors analyzed the only the adult song, it is not the first thing that comes to mind when asking how female calls emitted in response to juvenile songs might serve as an indicator of song quality. The current phrasing suggests that the authors would create a trajectory of song similarity over development and correlate this with a trajectory of female calling.

Lines 224-226: Can include just range of similarity scores here. (same with line 229).

Lines 226-229: include the finding that female call rate did not increase over time for these two juveniles.

Line 231: add reference.

Line 233: replace “feedback” with a different word, unjustified.

Line 234: “unrelated auditory feedback” – this is conjecture... just say that these juveniles did not experience an increase in calling by females over time.

Line 239: same comment as above about feedback.

Line 245-247: rephrase because these data do not demonstrate causality.

Lines 282-3030: were the same 15 units in 6 birds analyzed here? If so, please indicated in first or 2nd sentence of the paragraph.

Figure 3C. It would be useful to have the similarity of the pupil’s song to the tutor song above the panels (e.g., “Juvenile 1 (___%)”)

Figure 3D,E. Indicating what the horizontal lines mean on the figure itself would be useful (i.e., not just in legend).

line 266-270: given the author’s definition of spiking precision, one wonders why it is computed for silence.

Line 285: delete “to reflect the situation of auditory feedback”

Lines 305-323: I would appreciate it if the authors created a table that outlined how neurons changed their activity for each bird. For example, a table with five columns to reflect each bird and each row would indicate the number (and percentage) of neurons that increased their firing in response to call playback, how many neurons demonstrated decreases, and how many did not change. Or something else to better understand the data. This may all be moot because of the changes in analyses I suggest to harmonize approaches across juveniles and adults.

Figure 5D-F: these should be plotted as paired data (but see larger point about why these analyses are so different than the analyses of juvenile responses).

Line 355: what does n=22 call perturbations refer to? 22 exemplars of calls or 22 times a call was played back to the 9 projection neurons during singing? It seems like the latter, making it hard to image how statistics were done on this dataset.

Line 356-360: the overall experimental design and method of analysis are unclear here. For example, how is significance for each “case” analyzed?

Line 377-379: this statement should be qualified given the small sample size

Line 396-397: this sentence is awkward, I recommend rephrasing to emphasize how females help juveniles learn the tutor song.

Line 399: same comment as above about “genetic biases” actually reflecting effects of experience.

Line 405-406. The paper cited here is a paper on humans, but the wording makes it sound like the authors are talking about songbirds. In addition, the existing experiments on the role of social interactions in song tutor selection are also consistent with social effects on song memorization (e.g., experiments by Nicki Clayton), so the function of this sentence is unclear.

Line 406-409: “juvenile song learning” can refer to sensory or sensorimotor learning so the authors should be more specific here.

Line 406-409: While I believe that female influences on sensory learning should not be ruled out, I suggest that the authors can mention that juveniles practice A LOT during development and if females are providing reinforcement signals then there are more opportunities for females to shape sensorimotor learning than for them to shape sensory learning (in their experimental design at least).

Line 410: I think this heading is too strong and think that the following is more accurate “Female vocalizations during song practice could serve as social feedback” (or “Females COULD provide vocal feedback during song practice”). As mentioned in previous reviews, explicit experimentation to assess the degree to which female calls serve as social feedback (e.g., contingent vs. yoked calling playback) are important to conduct in order to make the existing statement. But I think it’s fair for the authors to speculate that female calls could serve as social feedback.

Lines 439-453: as indicating in my comments about statistical analyses, I think the current conclusions about the sensitivity of HVC projection neurons to female calls during singing are not supported as they currently stand. For example, it seems like the authors would NOT find an effect of female calls in juveniles if they conducted the same types of analyses used to analyze call-related signals in adult finches.

Line 500-504: It might be clearer to say that the “first syllable of the pair” compared to the “first and third syllable” when the previous sentence indicates that there are only two syllables in the tutor stimulus.

Line 518-522: this formula will compute the average syllable duration not the syllable rate (i.e., number of syllables per second)

Line 615-627: more details on data collection and analyses are required here. Right now, these experiments are very opaque. And given the importance of this experiment to the overall narrative, the details of this experiment need to be well described (e.g., how the perturbed trial was compared to control trials, how many perturbations per bird, were the same parts of song targeted?) It would be useful to know if the effect of the female call playback depended on where in the song the call was played (e.g., during a syllable or during a silent gap) but those sets of analyses might be outside the scope of the paper.

Reviewer #2 (Remarks to the Author):

The authors have addressed my concerns and I do not have any other comments. Except, I suggest that the authors use a chi-square analysis for the data in Fig 4F,L in order to substantiate this claim: 'Seven cells received stronger excitatory inputs during

300 the female call and eight received an increase in inhibitory inputs (Figure 4L). These

301 findings suggest that auditory responses to female calls in HVC are reduced during

302 female calls within BOS playback, when compared to auditory responses to female

303 calls alone (Figure 4F).'

I did a chi-square on their data and the p-value is 0.02, would be good if this was reported.

Reviewer #3 (Remarks to the Author):

The authors have done an excellent job in revising their manuscript. Thanks to the extended dataset, the study is now even stronger than it has been before. I am confident that the topic and the findings will be appreciated by a broad audience. I therefore recommend that the manuscript be accepted for publication in Nature Communications.

Reviewer #1 (Remarks to the Author):

I commend the authors for their additions and improvements to the manuscript. The analyses of similarity to father song, the addition of call analyses of two more birds housed with females, and the collection of more data from HVC projection neurons in juveniles have strengthened the manuscript. I continue to think that the authors are addressing an important and exciting issue in vocal learning.

We are grateful for Reviewer #1's acknowledgment of the manuscript's improvement after our initial revision.

However, there continue to be concerns about analyses and quality of data. For example, it is unclear why the authors use very different types of analyses for similar types of neurophysiological data for juvenile and adult birds.

While we agree that our data analysis is open for discussion, we wish to contest any claim of insufficient data quality. Our experiments were conducted thoroughly, with meticulous recording of behavioral data and electrophysiological data exhibiting an excellent signal-to-noise ratio. In addition, we would like to highlight once more that we successfully attained the first intracellular recordings in awake juvenile zebra finches during listening and singing which is a technically highly challenging achievement.

The variations in analysis methods can be attributed, in part, to differences in recording techniques. Specifically, intracellular recordings were exclusively employed for obtaining neural data in juveniles, while the inclusion of adult neural data involved extracellular recordings using Neuropixels probes. This approach was chosen to facilitate the assessment of a larger population of cells, a task not practically achievable with intracellular recordings. Furthermore, it is important to note that intracellular data from adult zebra finches under comparable conditions already existed (Vallentin & Long 2015), rendering their inclusion redundant and without additional value.

However, we agree with Reviewer #1 that the analysis of all datasets could be conducted more symmetrically, and we now streamlined our analysis accordingly (details below).

I realize that this is a substantive set of comments, but addressing these concerns will increase the rigor of their analyses, instill greater confidence in the quality of the data and interpretations, and ultimately strengthen their exciting paper.

We are thankful for the numerous valuable suggestions provided by Reviewer #1 to enhance the rigor of our analysis. Following the implementation of these recommendations, we affirm that the major results of our experiments remain robust and can be reported with confidence.

MAJOR COMMENTS:

Analysis of neural responses to female calls during song production:

- After conducting experiments designed to reveal the importance of female calls to juvenile song plasticity, the authors aim to identify neural populations that are sensitive to female calls during song practice. Overall, the data on the effect of call playback during song production are interesting but not compelling. And the authors overinflate the results. For almost 70% (15 out of 22) of the data, call playback leads to no change in the number of spikes.

We agree with Reviewer #1 that the neural changes in premotor neurons during female call perturbations in singing juveniles are interesting. We would like to assert that the significance of these changes remains compelling, even considering that not all instances are affected. Previous literature has reported and replicated that external auditory stimuli during listening and song production have no effect on the ongoing activity of premotor neurons in adults (Vallentin & Long 2015, Schmidt & Konishi 1998). To further highlight our findings, we now make a more direct comparison with the previously reported data set in adult birds (Vallentin & Long 2015) in respect to effect in spiking and subthreshold activity.

A linear mixed-effect model was calculated (details see below - Supplementary Table 1) and we confirm the Reviewer #1 's notion that on a population level the neuronal activity remains unaffected by the female call. However, we now compare the juvenile data with data recorded in adult birds. Since a female call during song production in adults did not alter the number of spikes in projection neurons we would like to highlight this difference and consider this as an effect in a subset of neurons that can only occur in juveniles. This result is now reported in line 347-356:

To explore if the neural activity of the projection neurons is affected by the female call, we presented a female call playback (n = 22 call perturbations) during 22 instances during 44 produced songs. On a population level, the neurons exhibited a similar number of spikes during perturbed and unperturbed syllables (linear mixed-effect model, mean number of spikes = 0.04 ± 0.40 , $p=0.912$, Figure 6D,F). However, in six out of the 22 cases when a female call occurred, neurons changed the number of spikes compared to baseline firing during song production (Figure 6D-F). This is an observation that is in contrast to data from adult zebra finches. In adults the number of spikes in projection neurons was unaffected when a female called during song production (Δ number of spikes=0; Figure 6F)⁴⁰.

To more directly compare the subthreshold activity of adult HVC projection neurons during female perturbations with the juvenile ones we considered the maximal delta subthreshold (as reported in Fig 3 in Vallentin and Long 2015) while a female called during singing in adult zebra finches as a threshold to determine whether a change in activity occurred. This threshold is now indicated in Fig. 6G as a solid black line and explained in the Figure legend. Therefore, subthreshold changes beyond this threshold could be considered as different from what has been observed in projection neurons in adults. This new direct comparison is now stated in line 356-363 and highlighted in Fig 6G:

To compare subthreshold changes caused by female call perturbations in adult and juvenile HVC projection neurons, we used the maximum delta subthreshold (as shown in Fig 3, Vallentin and Long 2015) during a female's call while an adult zebra finch was singing as a threshold to determine if any activity changes occurred (Figure 6G). We found that in 5 out of the 22 cases neurons exhibited subthreshold changes in response to female calls that extended beyond the maximally occurring changes observed in adult birds.

Additionally, in Figure 6F, the authors plot the absolute value of changes in spiking rates; when analyzing any change, plotting the absolute value can overestimate the effect. For example, if half the neurons increased in spike rate but the other half decreased in spike rate, then one would say that there wasn't a significant change in spiking activity (similar to

other analyses in this paper); but if you look at the absolute change, the effect is going to look large.

We apologize for seemingly inflating the observed effect. Our intention was to report a change and not to overestimate the effect. We now plotted this change as a relative change (Figure 6C) and any change that exceeded a 95% confidence interval was considered to be significant. Additionally, this effect was contrasted with absent changes in spiking activity in singing adult zebra finches during female call perturbations to further highlight the difference (line 347-356 and Figure 6F).

The authors need to adopt a more rigorous approach for this analysis.

- Similarly, since the authors emphasize differences in neural responses to female calls in adults (Vallentin and Long, 2015) and juveniles (this paper), it is important to conduct the same types of analyses across age groups. For example, one wonders what the analyses of adult responses would look like if the adult data were analyzed in precisely the same manner as analyzed here; my guess is that if you plot the absolute difference between control and experimental conditions in adults, you will likely have distributions of data that are above zero. More realistically, the authors should analyze juvenile responses in as similar a manner as possible to the adult data.

We thank Reviewer #1 for this suggestion. As described in our comments above we now conduct the same analysis for the data recorded in juveniles as reported for adult data (see Vallentin & Long 2015). This allows us to directly compare our results and underlines the effect of female calls during song production in juvenile birds but not adults (line 356-363 and updated Figure 6F,G).

- More details about data collection and analyses for this experiment are required in the Methods. Given the narrative of the paper, this is a very important experiment, and readers should be able to understand the details of the approaches. (In the response to reviewers file the authors write that “in 6 out of 22 female call perturbations the neural activity significantly changed compared to regularly occurring activity when the bird is singing without female call interruption (Figure 6F)” but it is not clear how statistical significance is computed for each of the 6 “female call perturbations”.)

We have added more details in line 347-356 and in the method section (line 691-700).

Analysis of neural responses to female calls in awake but quiet adult and juvenile finches: It is not clear why the analyses of auditory responses to female calls in awake but quiet adult finches and juvenile finches are so different. These are the same types of data.

In many respects, the analyses of HVC projection neurons in juveniles are more intuitive, whereas the analysis of HVC neurons in adults seems to be tailored to highlight neural responses. As far as I can tell, there is no reason why the adult data cannot be analyzed in the same manner as juvenile data, especially in the same paper.

We sincerely apologize for any confusion. A subthreshold analysis for adult data is absent due to the extracellular recording method. While firing rate and spiking precision were analyzed similarly for both juveniles and adults, we acknowledge that the presentation of the data lacked consistency, as pointed out by Reviewer #1. We appreciate this feedback and ensured uniform illustration of the data to address this concern. We have adjusted Figure 5 to present the data in a more symmetrical manner, aligning it with the format of Figure 4. We also streamlined the data analysis and now use linear mixed effect models to test for statistical significance (see Supplementary Table 1) and updated results section (juvenile data: line 267-277,282-284,295-299; adult data: line 312-319)

Analysis of the contribution of female calls to song improvement:

I appreciate the addition of two birds in the section on female vocal feedback increasing over development; these birds certainly help support the previous data. I also understand how this analysis helps segue into the neurophysiological experiments, but I continue to think that this experiment is preliminary. For example, only three of the five juveniles benefited from the presence of the female, and the juvenile that experienced the largest increase in call ratio (juvenile 3) did not produce a song that imitated the tutor song well. In a high profile journal like this, a strong statement about the impact of female calls during song practice on song learning (e.g., in the Abstract) that is based on $n=5$ will become amplified for people who do not read the details of the paper. While I believe that the best solution to strengthen this paper for this journal is to continue to increase the sample size (e.g., $n=8-10$ juveniles paired with females that increase call rates over time vs. $n=8-10$ juveniles paired with females that do not increase call rates over time), one possibility is to reword their abstract and discussion in a manner that highlights the preliminary nature of the analyses.

We appreciate the feedback from Reviewer #1. While we respectfully disagree on the need for an increase in the number of juveniles raised with females, we acknowledge the importance of addressing potential confounding factors. The substantial dataset, comprising 18672 calls ($n=4$ females) in response to 98344 songs ($n=5$ males), forms the basis of our conclusions. However, we agree that our experiment does not rule out the possibility of other external factors influencing song learning in juveniles. In response, we revised the wording in the abstract (line 37-40), the result section (line 251) and the discussion (line 380,402,415) to more carefully convey this aspect of our results.

Statistical models:

This should have been noted in the original submission but I realize that there are a number of places where statistics are inappropriate. My largest concern deals with the analysis of neurophysiological data. There is an increasing realization that statistics that treat each neuron within an individual as an independent sample are flawed (e.g., Yu et al., *Neuron*, 2022; Reed and Kaas, *Neural Networks*, 2010; Koerner and Zhang, *Brain Sciences*, 2017) and that mixed effects models are more appropriate for this type of data. To this end, the authors should analyze their data with some random factor; for example, bird ID should be added as a random factor in their statistical models for neurophysiological data. Right now, there is pseudoreplication in their analyses. For example, there is pseudoreplication when treating all 133 neurons in 5 birds as independent (Figure 5) as well as when treating the response of the same HVC neuron to multiple call perturbations (Figure 6). In the analysis of HVC responses to female calls during song production, it is possible that all six neurons that demonstrate changes in spike rate in response to call playback came from the same individual or even the same neuron (see comment above about the ambiguity of these analyses), and the current statistical approach does not account for this.

We appreciate the valuable suggestion from Reviewer #1. Upon careful consideration, we have re-evaluated all our statistical analyses and replaced the previous statistics with linear mixed-effect models where appropriate and feasible. Through this rigorous re-examination, our major results remained consistent. Importantly, we can now confidently assert that the reported effects are not attributed to individual birds or neurons. We thank the Reviewer #1 for guiding us towards this enhancement in our analysis, strengthening the robustness of our findings.

In addition, in response to a prior request from Reviewer #1, we enhanced our dataset by incorporating additional neuronal recordings (line 267-290 + adjusted the subpanels C,D,E, F of Figure 4). We then updated all statistical assessment in the main text and included a description of the statistical analysis in the method section (see below). We also provide a Supplementary Table 1 with more detailed results from these analyses relating them to the specific Figure subpanels.

(line 701-717):

Statistical analysis:

- Neuronal data: For metrics that could be measured in individual trials (firing rate), we analyzed the data for each trial, controlling for neuronal identity in order to account for any variability in the number of trials for each neuron. In metrics that measured properties across trials (precision), we analyzed the summary parameters for each neuron, controlling for bird identity. In order to account for the nested structure (neurons within birds) in continuous data, we fitted linear mixed model by maximum likelihood estimation (changes in neuron firing rate/spiking precision depending on condition (baseline or female call). We used the following model specifications:

neural activity $\sim 1 + \text{condition} + (1 | \text{bird ID}) + (1 | \text{bird ID:cell ID})$

- Behavioral data: To test for song similarity/similarity in syllable rate differences and account for the genetic background or early sensory experience of individual birds, we fitted a linear mixed model with the following model specifications:

Similarity (song or syllable rate) $\sim 1 + \text{condition} + (1 | \text{father ID}) + (1 | \text{father ID:bird ID})$.

Detailed results for individual statistical analysis can be found in the Supplementary Table 1.

Here I list various comments about statistical analyses, be it major or minor:

- lines 150-168: a within-individual analysis (not a two-sample t-test) should be used because each tutored bird has a similarity to the tutor and to the father. (It is possible that they actually ran a paired analysis for song similarity because they run a paired analysis for syllable rate later in the paragraph.) Because different birds can share a father and because genetics (or experience with a particular father) could affect learning, a mixed effects model with father ID as a random factor would be appropriate to analyze these data (similarity and syllable rate).

We agree with Reviewer #1 that a within-individual analysis should have been used. To test for song similarity differences and account for the genetic background or early sensory experience of individual birds, we now fitted a linear mixed-effect model (see Method section line 701-717 and Supplementary Table1). The results of this analysis are described in the result section (line 159-163).

- After reading the analyses in lines 156-160, one expects a comparison of similarities in syllable rates to the playback stimulus vs. to the father's song (e.g., pupil syllable rate compared to playback and father's syllable rates). This is a minor suggestion but I do I think this analysis would be a useful addition.

As per the recommendation, we have revised Figure 2 to distinctly illustrate the syllable rates of both the tutor song playback and the father's song (Figure 2C).

- Lines 144-147: indicating that they are analyzing the distribution of differences here (if different from zero) would be useful (but see larger comment about mixed models)

We appreciate the feedback from Reviewer #1, suggesting the application of linear mixed-effect models for analyzing our behavioral and neural data. While we have successfully implemented such models for electrophysiological data and behavioral experiments involving the comparison to fathers' songs, we would like to propose the use of non-parametric tests

for analyzing all data sets from Figure 1. This preference stems from the nature of the experiments where the data is not repeated; for instance, each bird can only have one song similarity score to the tutor song. Therefore, there are no random effects, such as bird identity, that need to be controlled for in these specific analyses. However we now adjusted the structure of the text to clarify our analysis (line 131-134).

Interestingly, birds copied the syllable rate of the tutor song playback (5.87 syll/sec) when raised with a female (5.44±1.17 syll/sec, p=0.137, one sample t-test) but not when raised alone (7.37±1.91 syll/sec, p=0.0091, one sample t-test).

- Lines 152-154: Justify why only a subset of juveniles are included in this analysis here. It seems like 5 out of 33 birds (15%) were excluded from this analysis, and it would be useful to briefly know why in the Results section.

We apologize for the oversight and would like to clarify the reason behind our inability to analyze the complete dataset. The absence of father's song recordings for seventeen birds prevented us from including them in the analysis. We have now revised the text to provide specific details and address any ambiguities (line 157-159):

We then compared the learned songs of a subset of juvenile birds with known father identities (n=16 birds) to the songs of their genetic fathers (Figure 2A, n=11 father birds).

- Lines 235-237 and lines 243-244: why are there comparisons between individual birds?

We now do not compare every single bird but develop a criterion that allows us to determine which bird was a good versus poor learner in respect to copying the tutor song (line 240-246):

We employed the similarity between untutored birds' songs, along with a 95% confidence interval as a threshold, to categorize birds into good and poor learners based on their ability to replicate the provided tutor song. Scrutinizing this criterion (40.7% similarity threshold) led to the observation that juvenile birds, receiving progressively more auditory input from females regarding their song performance, exhibited better replication of the tutor song compared to juveniles that did not experience an increase in calling by females over time (Figure 3D).

- Line 300-303: The authors can explicitly compare the degrees of change using a paired analysis (mixed effects model) instead of just making a qualitative statement.

We incorporated this comment and now compared the degrees of change (line 284-290 and line 300-301 and the Figure legend of Figure 4 K,L).

(line 284-290):

The overall change in subthreshold input during female call presentation was not different from regular occurring subthreshold activity during silence (median Δ subthreshold= 1.5835 mV, p=0.56, Wilcoxon rank sum test) (Figure 4F) indicating that the more correlated activity observed in the subthreshold activity (Figure 4E) is not systematic but either reflects a release from local inhibition or the transmission of excitatory female call-related information from upstream auditory areas.

line (300-301):

In the BOS + female call condition, nor the subthreshold correlation neither the change in subthreshold activity were significantly different on a population level (Figure 4K,L).

Song similarity:

Line 514-517. This calculation escaped me on my first read through. Is there a strong justification for this formula? The vast majority of studies using SAP to quantify song learning simply by analyzing the similarity of BOS to tutor song (i.e., one of the variables in the

numerator). It is not clear why the SAP similarity score is multiplied with the accuracy score and then normalized for these analyses, so a justification is important here. Because I suspect that other readers will similarly interpret the data in their figures in the same manner as I originally did, the authors should use a different name than “song similarity” in the main text and figures so people do not try comparing the song similarity scores in this paper to song similarity scores in other papers. In addition, given the prevalence of analyses of SAP similarity scores, it would be very useful to analyze and plot this data in a supplementary figure (or in the main text).

We appreciate Reviewer #1 for raising this point. Previously, we computed song similarity following the described approach, as outlined in Kosche et al. (2015) and Vallentin et al. (2016), as recommended by Ofer Tchernichovski for assessing 'Global similarity' (personal communication). However, in response to Reviewer #1's suggestion, we acknowledge the suitability of utilizing 'Local similarity' for our objectives. Subsequently, we have reanalyzed the similarity scores presented in Figure 1, Figure 2, Figure 3 and Supplementary Figure 3. The results remain consistent with our previous findings, albeit with slightly increased similarity values.

MINOR COMMENTS:

Abstract: line 40: I suggest replacing “as young birds” with “the males that they were housed with” because the “young birds” in the current wording could refer to the females themselves.
Changed.

Abstract: line 40: replace “indicating” with “suggesting” since the former is too strong.
Changed.

Line 44-49: “song-related” could refer to listening to song, so “singing-related” is better. Please replace.
Changed.

Line 80: why limit this to “their offspring”? Are you specifically talking about mothers (that is not evident here)? “their offspring” can be replaced with “males” or “developing males”
Changed.

Lines 117-119: can the authors say that the tutor song was acoustically distinct from the father's song? This would be useful to say (depending on the degree to which this is true) for experiments on juveniles that were previously tutored by their fathers.
We now compared each individual father song with the tutor song and the resulting distribution had a mean of 31.45% which is comparable to comparing untutored birds to the tutor song (untutored birds' similarity to tutor song = 32.02%, $p=0.833$, Wilcoxon rank sum test). This is now mentioned in the text (line 153-157) and added to Figure legend of Figure 2B).

Line 126: Suggest adding a paragraph break here to make the results in lines 121-126 more impactful. Right now this paragraph is very long, and important results can get buried.
Changed.

Lines 144-148: it seems more impactful to put these results BEFORE the results on syllable and gap durations. After reading the results on syllable durations, I wondered why these results were presented before the results on syllable rate (which are more compelling for learning).
Changed.

Figure 1A legend: add sample sizes for each group in figure legend. This would generally be useful in legends throughout the manuscript.

Added.

Line 150-154: I wouldn't call these genetic predispositions, since they are based on experiences during development. Moreover, the authors should explicitly mention in the Results section that the juveniles were raised with fathers for some period of time and could have learned their father's song to some degree.

Changed.

Figure 2A: It is hard to see which tutored bird is tutored with or without females. Maybe the authors could have orange vs. green backgrounds behind the panels (or a vertical orange or green line to the left of the spectrograms) to make the difference between birds more conspicuous.

Changed.

Figure 2B: add label for which birds were tutored with or without females above the figure to make the figure easier to understand.

Changed.

Lines 183-184: same comment about genetic predispositions rephrased as experience-dependent learning.

Changed.

Lines 202-207: in parentheses, just list range without listing numbers per bird. This is one of the many places where there is too much text in parentheses.

Wherever possible, we reduced the text in parentheses throughout the manuscript.

Line 211: replace "learning" with "performance". The former requires that they analyze similarity to tutor song throughout development (which isn't a bad idea, but probably quite labor intensive).

Changed.

line 214-217: awkward sentence, unclear how juvenile 5 fits into this or why other juveniles that were raised with different females were not compared here. For the reader to understand the full extent of these comparisons, the authors should first indicate which juveniles were housed with the same females. Or just simplify this sentence.

Changed.

Line 220-222: I suggest rewording these sentences (or at least the first of these two). While I understand why the authors analyzed the only the adult song, it is not the first thing that comes to mind when asking how female calls emitted in response to juvenile songs might serve as an indicator of song quality. The current phrasing suggests that the authors would create a trajectory of song similarity over development and correlate this with a trajectory of female calling.

Changed and added additional clarification on how we classified good versus poor learners (line 240-246).

Lines 224-226: Can include just range of similarity scores here. (same with line 229).

Changed.

Lines 226-229: include the finding that female call rate did not increase over time for these two juveniles.

Changed.

Line 231: add reference.
Deleted and data added instead.

Line 233: replace “feedback” with a different word, unjustified.
Changed.

Line 234: “unrelated auditory feedback” – this is conjecture... just say that these juveniles did not experience an increase in calling by females over time.
Changed.

Line 239: same comment as above about feedback.
Changed.

Line 245-247: rephrase because these data do not demonstrate causality.
Changed.

Lines 282-3030: were the same 15 units in 6 birds analyzed here? If so, please indicated in first or 2nd sentence of the paragraph.
We have added a clarification (line 294).

Figure 3C. It would be useful to have the similarity of the pupil’s song to the tutor song above the panels (e.g., “Juvenile 1 (__%)”)
Changed.

Figure 3D,E. Indicating what the horizontal lines mean on the figure itself would be useful (i.e., not just in legend).
Changed.

line 266-270: given the author’s definition of spiking precision, one wonders why it is computed for silence.
We provide a justification for why spiking precision is also calculated during a baseline silent period in line 271-273:
Due to the variability in the firing properties of each cell, we initially computed spiking precision during periods of silence as a baseline.

Line 285: delete “to reflect the situation of auditory feedback”
Deleted.

Lines 305-323: I would appreciate it if the authors created a table that outlined how neurons changed their activity for each bird. For example, a table with five columns to reflect each bird and each row would indicate the number (and percentage) of neurons that increased their firing in response to call playback, how many neurons demonstrated decreases, and how many did not change. Or something else to better understand the data. This may all be moot because of the changes in analyses I suggest to harmonize approaches across juveniles and adults.

We now streamlined our analysis across data sets and provide a clear overview about the statistics that were used. We now provide a table including the statistics applied. The data will be made available upon publication.

Figure 5D-F: these should be plotted as paired data (but see larger point about why these analyses are so different than the analyses of juvenile responses).
Changed.

Line 355: what does n=22 call perturbations refer to? 22 exemplars of calls or 22 times a call

was played back to the 9 projection neurons during singing? It seems like the latter, making it hard to image how statistics were done on this dataset.

Changed (see above)

Line 356-360: the overall experimental design and method of analysis are unclear here. For example, how is significance for each “case” analyzed?

See comment above.

Line 377-379: this statement should be qualified given the small sample size

To accurately represent the data, the sentence was modified.

Line 396-397: this sentence is awkward, I recommend rephrasing to emphasize how females help juveniles learn the tutor song.

Changed.

Line 399: same comment as above about “genetic biases” actually reflecting effects of experience.

Changed.

Line 405-406. The paper cited here is a paper on humans, but the wording makes it sound like the authors are talking about songbirds. In addition, the existing experiments on the role of social interactions in song tutor selection are also consistent with social effects on song memorization (e.g., experiments by Nicki Clayton), so the function of this sentence is unclear.

Changed.

Line 406-409: “juvenile song learning” can refer to sensory or sensorimotor learning so the authors should be more specific here.

Changed.

Line 406-409: While I believe that female influences on sensory learning should not be ruled out, I suggest that the authors can mention that juveniles practice A LOT during development and if females are providing reinforcement signals then there are more opportunities for females to shape sensorimotor learning than for them to shape sensory learning (in their experimental design at least).

A statement about the potential of female calls supporting sensory learning was added in line 433-435:

Given the potential of female calls to serve as a rewarding stimulus it should be further explored whether and how female calls also shape sensory learning of the tutor song.

Line 410: I think this heading is too strong and think that the following is more accurate “Female vocalizations during song practice could serve as social feedback” (or “Females COULD provide vocal feedback during song practice”). As mentioned in previous reviews, explicit experimentation to assess the degree to which female calls serve as social feedback (e.g., contingent vs. yoked calling playback) are important to conduct in order to make the existing statement. But I think it’s fair for the authors to speculate that female calls could serve as social feedback.

Changed.

Lines 439-453: as indicating in my comments about statistical analyses, I think the current conclusions about the sensitivity of HVC projection neurons to female calls during singing are not supported as they currently stand. For example, it seems like the authors would NOT find an effect of female calls in juveniles if they conducted the same types of analyses used to analyze call-related signals in adult finches.

We have now included a direct comparison of adult and juvenile data, further reinforcing our earlier statement.

Line 500-504: It might be clearer to say that the “first syllable of the pair” compared to the “first and third syllable” when the previous sentence indicates that there are only two syllables in the tutor stimulus.

Changed.

Line 518-522: this formula will compute the average syllable duration not the syllable rate (i.e., number of syllables per second)

Changed.

Line 615-627: more details on data collection and analyses are required here. Right now, these experiments are very opaque. And given the importance of this experiment to the overall narrative, the details of this experiment need to be well described (e.g., how the perturbed trial was compared to control trials, how many perturbations per bird, were the same parts of song targeted?) It would be useful to know if the effect of the female call playback depended on where in the song the call was played (e.g., during a syllable or during a silent gap) but those sets of analyses might be outside the scope of the paper. More details are provided in the text (line 347-354) and method section (line 691-700).

Reviewer #2 (Remarks to the Author):

The authors have addressed my concerns and I do not have any other comments. Except, I suggest that the authors use a chi-square analysis for the data in Fig 4F,L in order to substantiate this claim: 'Seven cells received stronger excitatory inputs during 300 the female call and eight received an increase in inhibitory inputs (Figure 4L). These 301 findings suggest that auditory responses to female calls in HVC are reduced during 302 female calls within BOS playback, when compared to auditory responses to female 303 calls alone (Figure 4F).'

I did a chi-square on their data and the p-value is 0.02, would be good if this was reported.

We thank Reviewer '2 for this suggestion. Based on Reviewer # 1's suggestions we now re-analyzed these data with linear mixed effect models.

Reviewer #3 (Remarks to the Author):

The authors have done an excellent job in revising their manuscript. Thanks to the extended dataset, the study is now even stronger than it has been before. I am confident that the topic and the findings will be appreciated by a broad audience. I therefore recommend that the manuscript be accepted for publication in Nature Communications.

We thank Reviewer #3 for the appreciation of our work.

REVIEWER COMMENTS

Reviewer #1 (Remarks to the Author):

General comments:

I sincerely thank the authors for the various changes to the analyses, the inclusion of additional methodological details, and the modification of wording throughout. For example, I appreciate the more explicit comparisons to published data in adult finches (see analyses related to figure 6), the analysis of %similarity SAP scores (to make this more comparable to other published papers), and the inclusion of important information of developmental experiences of birds. The manuscript is much improved. However, I continue to think that the key experiment of the manuscript - the recording of HVC responses to female calls during juvenile singing – remains too preliminary to make general statements. I first describe my continued concerns about this experiment (and need for additional information) and then describe various other comments and requests for clarification.

Sensitivity of juvenile HVC neurons to the sounds of female calls:

All in all, the additional analyses included in the revised manuscript give me more assurance about the findings, and I think it is possible that HVC neurons in juvenile zebra finches are responsive to female calls. I appreciate the difficulty of these recordings and experiments but I continue to think that there is insufficient data to make strong and broad statements about the sensitivity of HVC neurons in juveniles to female calls. And if the existing dataset is ultimately published (in this journal or in other journals), the authors should indicate that these data are preliminary (see summary comments at the end of this section).

1. “On a population level, the neurons exhibited a similar number of spikes during perturbed and unperturbed syllables (mean number of spikes = 0.04 ± 0.40 , $p=0.917$, linear mixed-effect model Figure 6D, F). However, in six out of the 22 cases when a female call occurred, neurons changed the number of spikes compared to baseline firing during song production (Figure 6D-F). This is an observation that is in contrast to data from adult zebra finches. In adults the number of spikes in projection neurons was unaffected when a female called during song production (Δ number of spikes=0; Figure 6F).”

1a. First, please indicate which figure in Vallentin and Long 2015 depicts “ Δ number of spikes” in adult finches in response to call playback during song production, and include the number of neurons used to compute this (so readers don’t have to go to that paper to look it up). It would be useful to know since I didn’t see these exact data in Vallentin and Long 2015 (e.g., differences in spiking activity in response to deafening was reported but I didn’t see analyses of spiking in response to calls during song). I appreciate that the authors specifically referenced a figure in their previous published paper regarding subthreshold responses (“Figure 3 of Vallentin and Long 2015”).

1b. More information is needed to understand how all the adult data were computed. For example, is there any variance associated with “ Δ number of spikes=0” in adults? Is it the case that there was never a change in spikes, and if so, how many neurons were analyzed. If there is variation in the change in spiking activity across adult HVC neurons, then the wording (and conclusions stemming from this wording) could be misleading because the authors are comparing 6 instances of responses to the population mean in adults. A very sensible analysis would be to compare population changes in juveniles and in adults, and it is highly likely that this difference would not be significantly different.

1c. If it is the case that every adult HVC neuron did not demonstrate a change in spikes in response to female call playback, then the current description and analysis seems more valid. As mentioned above, if this is indeed the case, the authors should describe the data in adults more comprehensively (number of neurons in how many birds in response to how many instances of perturbations).

1d The fact that there was no significant effect of female calls on HVC activity at a population level is important to keep in the manuscript and supports my apprehension in making strong statements about the sensitivity of juvenile HVC neurons to female call.

2. “We found that in 5 out of the 22 cases neurons exhibited subthreshold changes in response to female calls that extended beyond the maximally occurring changes observed in adult birds.”

2a. As far as I can tell, the neurons analyzed here cannot be specifically classified as RA- or X-projecting HVC neurons. The new analyses include a comparison of the individual responses of HVC projection neurons in juveniles to the maximum changes to activity in adults (Vallentin and Long, 2015). Upon close examination, the maximum threshold (in adults) used for this comparison seem to be specifically for HVC-RA neurons (black lines; ~ 1 mV); the maximum threshold for HVC-X neurons is considerably higher, closer to 1.5mV (see bottom right panel of Figure 3D in Vallentin and Long, 2015). It is very possible that the results continue to hold with this higher threshold. But given the unknown population of projection neurons that they are recording from, the authors need to compare the current data with both thresholds or just the higher threshold in the manuscript.

2b. Upon closer inspection, is the red line in Figure 6G the threshold for X-projecting neurons in the adult HVC? This red line is not mentioned in the text or figure legend. If so, then the authors just need to change the figure legend as mentioned above.

3. The authors should indicate how many neurons these six instances (Figure 6F) were from. It would be concerning if all instances came from a single neuron or from a single bird.

4. The data so far indicate that there was a change in HVC activity on six instances (out of 22 instances) of female call playback (five of which seem to be larger than the biggest responses in adults). To go from these data to a broad statement that HVC projection neurons in juveniles are responsive to female calls is inappropriate.

- Depending on how the authors respond to the queries above (e.g., if the six instances of changes to HVC activity are from just one neuron or just one bird), I propose three solutions for going forward. First, the authors could collect more data to add greater support for making a broad statement about HVC neurons. Second, the authors could remove this section of the manuscript

because of the very small sample size of instances with changes in activity and the danger of generalizing from these few examples (as well as the lack of change on a population level). This would allow this data to be published elsewhere when more data are collected. Third, the authors could phrase the Abstract, Results, and Discussion in ways that clearly outline the preliminary nature of the findings. Given that many people just read the abstract, ensuring that people understand the limited scope of the data in the Abstract is crucial. For example, one could replace lines 43-49 in the abstract with “In addition, we found preliminary evidence that HVC neurons of juvenile birds are sensitive to the sound of female calls when juveniles are engaged in vocal practice and that the magnitude of call-related signals in HVC in juveniles is qualitatively larger than those observed in adults.” But as I mentioned above, the wording needs to be changed throughout the manuscript.

Other comments:

- The use of the word “feedback”. I am completely fine with the idea that females can provide feedback to guide vocal learning in juveniles. But one needs to be rigorous about when to use this term, since it will lose its meaning if not used properly (i.e., if used without sufficient evidence). If females are providing feedback and reinforcing particular renditions of song, then the timing of their calls should be non-random and such targeted productions of female calls should guide the trajectory of song learning. And, as the authors now write in the Discussion, it is important to test whether female vocalizations serve as feedback by conducting contingent vs. non-contingent playback experiments. However, without the data that supports describing female calls as feedback, the use of “feedback” in many of the places in this manuscript remains conjecture. In my previous reviews I have asked the authors to scale back on the use of feedback and there have been improvements in wording. But I will be very explicit now (hopefully for the last time) in where and how “feedback” should be changed given the current state of knowledge. For each of these instances, feedback should be replaced with “vocalizations”, “calling”, or something less infused with function.

- in the title

- but seems OK to keep “feedback” in the list of keywords

- lines 41, 50, 173, 204, 228, 249, 339, 377, 414, 456, 458, & 569

- (in the instances that I don’t recommend changing the wording, the authors frame their results as suggesting something akin to “feedback”)

- when submitting the revision, I urge the authors to be conservative in their use of “feedback” in new text.

- line 157-163: “We then compared the learned songs of a subset of juvenile birds with known father identities (n=16 birds) to the songs of their genetic fathers (Figure 2A, n=11 father birds). We observed that birds raised with a female had a higher song similarity to the tutor song and a better match of the tutor’s song syllable rate compared to the isolated ones (song similarity: $p=0.007$,

syllable rate: $p=0.003$, linear mixed-effect model, Figure 2A-C) independent of their genetic background or early sensory experience prior to 30 days post hatch.” I can understand the importance of analyzing only birds with recordings of their father song. But that restriction is needed only if the authors are explicitly analyzing the songs of the pupils to the songs of the father. The current wording of the results make it sound like the authors are simply comparing the songs of birds housed with females to the songs of birds housed individually. So, based on the current wording, there is no comparison to the father’s song, and there is no need to exclude the data for 17 pupils (since only 16 of the 33 birds are analyzed here). But maybe my concern reflects inaccurate wording in the text. When I look at Figure 2, I see that juvenile males housed with a female produced songs that were significantly more similar to the tutor’s song than to the father’s song. On the other hand, juveniles raised without females produced songs that bear equal resemblance to the tutor and father’s songs (if anything, their songs are more similar to the father’s song (i.e., there seems to be one bird that is driving the lack of significance)). If this is indeed what the authors intend to communicate, they should revise the wording here to more accurately reflect the analyses. The same applies to the description of the analysis of syllable rate.

- I thank the authors for modifying the equation pertaining to syllable rate but can the authors include a justification for this in the manuscript? Even now the formula seems odd to me. It’s basically the inverse of the average syllable duration (which was what was calculated in the previous submission), which is not the same as the canonical definition of syllable rate. For example, in Mets and Brainard 2019 (a paper that is cited by the authors ([44]), syllable rate (“tempo”) is calculated as “the number of syllables present in a song bout divided by the duration of the song bout”. In other words, this calculation includes the duration of gaps in the denominator to give a common measure of tempo (see also Leadbeater et al., 2005; Forstmeier et al., 2009; Rouse et al, 2023; etc.).

☒ The year of Mets and Brainard is 2019, not 2017.

- Figure 2, line 202: There are no “dashed” lines in the figure, just black lines

- Line 232: “female call ratio” is not defined before this, making it hard to understand. Please explicitly define.

- Legend for Figure 3D: indicate that this is the 95% CI (to harmonize with text).

- Line 284-286: why use this statistic? They should still use a mixed effects model since there are multiple neurons per bird. The distribution of differences looks like it could be different than zero, depending on which neurons belonged to which birds. I would make the same point for statistics related to Figure 4L and all other data.

- Line 294: delete “during song practice” because these juveniles are not singing.

- Line 302-304: The authors can explicitly analyze the strength of responses to female calls outside or during BOS playback because 15 neurons were recorded under both conditions. So if the authors really want to make this statement, they should conduct the analysis. They should not base this

statement on differences in the level of significance between panels C-F and G-L because sample sizes are larger in the analyses depicted in C-F (therefore more power).

- Legend for Figure 4L: add SE or SD.

- Legend for Figure 4: Generally speaking, while I appreciate the details about the data, if the authors are going to add details about means in the figures for the BOS conditions (G-L), they should do the same for the non-BOS condition (C-F) for consistency. But honestly, I don't think they need to report the means here.

- Line 310: add "extracellular" here to be clear and to avoid other readers from wanting an explicit comparison between juvenile and adult data. Could also add to figure 5 to avoid confusion.

- Figure 6G: I wouldn't use "BOS" here since the same acronym is used to refer to HVC activity during the playback of the bird's own song. Write "singing" or something that differentiate these data from the data in Figure 4.

- Line 383: Figure 5 should not be referenced since this figure summarizes data from adults.

- Line 418: "When female vocalizations were not correlated with the learning phase"... what is the learning phase? And what does it mean to not be correlated with it? Does it refer to when juveniles are practicing (singing), the developmental period of vocal practice, or something else?

- Line 551: DAS was defined only in a figure legend, and TCN has not been defined so please write out both here.

- Line 561-563: A little more clarity about the window of interest is important for people to replicate these experiments and analyses. Is the beginning of the window the onset of the first introductory note, and if so, what is the duration of silence used to differentiate if an introductory note is part of the song bout or not. Also, just to clarify, the end of the window is the offset of the last syllable of the bout 350 ms?

- Line 572-575: please clarify if the exact same part of song is extracted for the control condition? Is that what "at the same duration from song onset as the perturbed snippets" is supposed to indicate? It seems unclear to me from the current wording.

- Supplementary Figure 1: It seems like this bird didn't learn the tutor song so I would just say it "produced a" song that....

- Supplementary Figure 3: what does each data point represent? a single exemplar or the mean from a particular syllable type?

Point-by-point responses to the comments of Reviewer #1 (reviewers' comments in black, our responses in green) to the manuscript 'Female calls promote song learning in male juvenile zebra finches' by Linda Bistere, Carlos M. Gomez-Guzman, Yirong Xiong and Daniela Vallentin

Reviewer #1 (Remarks to the Author):

General comments:

I sincerely thank the authors for the various changes to the analyses, the inclusion of additional methodological details, and the modification of wording throughout. For example, I appreciate the more explicit comparisons to published data in adult finches (see analyses related to figure 6), the analysis of %similarity SAP scores (to make this more comparable to other published papers), and the inclusion of important information of developmental experiences of birds. The manuscript is much improved. However, I continue to think that the key experiment of the manuscript - the recording of HVC responses to female calls during juvenile singing – remains too preliminary to make general statements. I first describe my continued concerns about this experiment (and need for additional information) and then describe various other comments and requests for clarification.

We sincerely appreciate Reviewer #1's recognition of the improvements made to our manuscript. We apologize for any remaining lack of clarity and aim to thoroughly address the suggested revisions in this response. We are confident that we can satisfactorily resolve any outstanding concerns.

Sensitivity of juvenile HVC neurons to the sounds of female calls:

All in all, the additional analyses included in the revised manuscript give me more assurance about the findings, and I think it is possible that HVC neurons in juvenile zebra finches are responsive to female calls. I appreciate the difficulty of these recordings and experiments but I continue to think that there is insufficient data to make strong and broad statements about the sensitivity of HVC neurons in juveniles to female calls. And if the existing dataset is ultimately published (in this journal or in other journals), the authors should indicate that these data are preliminary (see summary comments at the end of this section).

We appreciate Reviewer #1 considering the possibility that HVC neurons in juveniles may be responsive to female calls. Our data, revealing a significant increase in firing rate when juveniles are exposed to female calls alone (Figure 3), strongly supports this hypothesis. We acknowledge the smaller dataset for singing juveniles, with only a subset of neurons displaying explicit firing pattern changes in response to female calls. However, we emphasize the absence of similar effects in the previous study on singing adults by the senior author (Vallentin & Long 2015). A thorough re-analysis of this data has been integrated into our study (lines 345-357), and we will further discuss this contrasting finding in our detailed response to the specific comments.

1. "On a population level, the neurons exhibited a similar number of spikes during perturbed and unperturbed syllables (mean number of spikes = 0.04 ± 0.40 , $p=0.917$, linear mixed-effect model Figure 6D, F). However, in six out of the 22 cases when a female call occurred, neurons changed the number of spikes compared to baseline firing during song production (Figure 6D-F). This is an observation that is in contrast to data from adult zebra finches. In adults the number of spikes in projection neurons was unaffected when a female called during song production (Δ number of spikes=0; Figure 6F)."

1a. First, please indicate which figure in Vallentin and Long 2015 depicts " Δ number of spikes" in adult finches in response to call playback during song production, and include the

number of neurons used to compute this (so readers don't have to go to that paper to look it up). It would be useful to know since I didn't see these exact data in Vallentin and Long 2015 (e.g., differences in spiking activity in response to deafening was reported but I didn't see analyses of spiking in response to calls during song). I appreciate that the authors specifically referenced a figure in their previous published paper regarding subthreshold responses ("Figure 3 of Vallentin and Long 2015").

We thank the Reviewer for this comment and acknowledge our imprecision in referencing the previously published dataset. Reviewer #1 is correct that the change in spike count within HVC neurons recorded in singing adult zebra finches in response to female calls was not explicitly reported in Vallentin and Long (2015). We have re-analyzed this dataset and now report the precise number of neurons recorded and perturbations analyzed (lines 345-357), To compare the adult data with the juvenile data, we performed a Fisher's Exact test on the number of instances where a female call occurred during song production. The test specifically examined whether there was a significant difference between juveniles and adults in the number of cases with a change (or no change) in the number of spikes compared to baseline firing (spiking activity during uninterrupted singing):

	adult	juvenile	
	———	———	
Number of cases when a change in # spikes occurred	0	6	
Number of cases when no change in # spikes occurred	98	16	p = 2.0427e-05

Alternatively, the Fisher's Exact test also yields a significant result when analyzing the number of neurons that displayed a change in spiking activity in response to female calls.

	adult	juvenile	
	———	———	
Number of neurons displaying a change in # spikes	0	4	
Number of neurons displaying a no change in # spikes	12	5	p = 0.0211

In conclusion, female calls during song production have no effect on the ongoing spiking activity in adult birds which is in stark contrast to what we observed in juveniles.

1b. More information is needed to understand how all the adult data were computed. For example, is there any variance associated with " Δ number of spikes=0" in adults? Is it the case that there was never a change in spikes, and if so, how many neurons were analyzed. If there is variation in the change in spiking activity across adult HVC neurons, then the wording (and conclusions stemming from this wording) could be misleading because the authors are comparing 6 instances of responses to the population mean in adults. A very sensible analysis would be to compare population changes in juveniles and in adults, and it is highly likely that this difference would not be significantly different.

1c. If it is the case that every adult HVC neuron did not demonstrate a change in spikes in response to female call playback, then the current description and analysis seems more valid. As mentioned above, if this is indeed the case, the authors should describe the data in adults more comprehensively (number of neurons in how many birds in response to how many instances of perturbations).

In response to the reviewer's insightful suggestions, we have revised our description and are confident that we have now comprehensively demonstrated the significance of the observed effect of female calls on singing in juveniles (lines 345-357).

1d The fact that there was no significant effect of female calls on HVC activity at a population level is important to keep in the manuscript and supports my apprehension in making strong statements about the sensitivity of juvenile HVC neurons to female call.

Despite our new comparison with adult data, we maintained the finding that there was no population-level effect in the juvenile dataset. We agree with Reviewer #1 that this effectively demonstrates that not all HVC neurons are sensitive to female calls, but rather a specific subset exhibits this sensitivity.

2. "We found that in 5 out of the 22 cases neurons exhibited subthreshold changes in response to female calls that extended beyond the maximally occurring changes observed in adult birds."

2a. As far as I can tell, the neurons analyzed here cannot be specifically classified as RA- or X-projecting HVC neurons. The new analyses include a comparison of the individual responses of HVC projection neurons in juveniles to the maximum changes to activity in adults (Vallentin and Long, 2015). Upon close examination, the maximum threshold (in adults) used for this comparison seem to be specifically for HVC-RA neurons (black lines; ~1 mV); the maximum threshold for HVC-X neurons is considerably higher, closer to 1.5mV (see bottom right panel of Figure 3D in Vallentin and Long, 2015). It is very possible that the results continue to hold with this higher threshold. But given the unknown population of projection neurons that they are recording from, the authors need to compare the current data with both thresholds or just the higher threshold in the manuscript.

2b. Upon closer inspection, is the red line in Figure 6G the threshold for X-projecting neurons in the adult HVC? This red line is not mentioned in the text or figure legend. If so, then the authors just need to change the figure legend as mentioned above.

We thank Reviewer #1 for this observation. We have adjusted the maximum threshold to incorporate the HVC-X neuron data from adult birds. Our previously reported results remain valid with this adjustment.

3. The authors should indicate how many neurons these six instances (Figure 6F) were from. It would be concerning if all instances came from a single neuron or from a single bird.

The six instances of altered spiking activity in response to female calls were observed across all four neurons that displayed a change, with each neuron originating from a different bird. This detail has been incorporated into the text (lines 345-357).

4. The data so far indicate that there was a change in HVC activity on six instances (out of 22 instances) of female call playback (five of which seem to be larger than the biggest responses in adults). To go from these data to a broad statement that HVC projection neurons in juveniles are responsive to female calls is inappropriate.

- Depending on how the authors respond to the queries above (e.g., if the six instances of changes to HVC activity are from just one neuron or just one bird), I propose three solutions for going forward. First, the authors could collect more data to add greater support for making a broad statement about HVC neurons. Second, the authors could remove this section of the manuscript because of the very small sample size of instances with changes

in activity and the danger of generalizing from these few examples (as well as the lack of change on a population level). This would allow this data to be published elsewhere when more data are collected. Third, the authors could phrase the Abstract, Results, and Discussion in ways that clearly outline the preliminary nature of the findings. Given that many people just read the abstract, ensuring that people understand the limited scope of the data in the Abstract is crucial. For example, one could replace lines 43-49 in the abstract with "In addition, we found preliminary evidence that HVC neurons of juvenile birds are sensitive to the sound of female calls when juveniles are engaged in vocal practice and that the magnitude of call-related signals in HVC in juveniles is qualitatively larger than those observed in adults." But as I mentioned above, the wording needs to be changed throughout the manuscript.

We appreciate Reviewer #1's suggestions and have clarified that only a subset of recorded HVC neurons is sensitive to female calls. Given the direct comparison of our juvenile data with a similarly sized adult dataset, which reveals a clear difference, we respectfully disagree that a larger juvenile dataset is necessary. We have revised the abstract (lines 43-46) to emphasize the contrast between our juvenile observations and existing adult data, providing evidence for the sensitivity of HVC neurons to female calls in juveniles. This change is also reflected in the revised Results (lines 345-357) and Discussion (lines 457-462) sections.

Other comments:

- The use of the word "feedback". I am completely fine with the idea that females can provide feedback to guide vocal learning in juveniles. But one needs to be rigorous about when to use this term, since it will lose its meaning if not used properly (i.e., if used without sufficient evidence). If females are providing feedback and reinforcing particular renditions of song, then the timing of their calls should be non-random and such targeted productions of female calls should guide the trajectory of song learning. And, as the authors now write in the Discussion, it is important to test whether female vocalizations serve as feedback by conducting contingent vs. non-contingent playback experiments. However, without the data that supports describing female calls as feedback, the use of "feedback" in many of the places in this manuscript remains conjecture. In my previous reviews I have asked the authors to scale back on the use of feedback and there have been improvements in wording. But I will be very explicit now (hopefully for the last time) in where and how "feedback" should be changed given the current state of knowledge. For each of these instances, feedback should be replaced with "vocalizations", "calling", or something less infused with function.

- in the title
- but seems OK to keep "feedback" in the list of keywords
- lines 41, 50, 173, 204, 228, 249, 339, 377, 414, 456, 458, & 569
- (in the instances that I don't recommend changing the wording, the authors frame their results as suggesting something akin to "feedback")
- when submitting the revision, I urge the authors to be conservative in their use of "feedback" in new text.

We have replaced the word "feedback" with alternative phrasing at the suggested instances.

- line 157-163: "We then compared the learned songs of a subset of juvenile birds with known father identities (n=16 birds) to the songs of their genetic fathers (Figure 2A, n=11 father birds). We observed that birds raised with a female had a higher song similarity to the tutor song and a better match of the tutor's song syllable rate compared to the isolated ones (song similarity: $p=0.007$, syllable rate: $p=0.003$, linear mixed-effect model, Figure 2A-C) independent of their genetic background or early sensory experience prior to 30 days post hatch." I can understand the importance of analyzing only birds with recordings of their father song. But that restriction is needed only if the authors are explicitly analyzing the songs of the pupils to the songs of the father. The current wording of the results make it sound like the

authors are simply comparing the songs of birds housed with females to the songs of birds housed individually. So, based on the current wording, there is no comparison to the father's song, and there is no need to exclude the data for 17 pupils (since only 16 of the 33 birds are analyzed here). But maybe my concern reflects inaccurate wording in the text. When I look at Figure 2, I see that juvenile males housed with a female produced songs that were significantly more similar to the tutor's song than to the father's song. On the other hand, juveniles raised without females produced songs that bear equal resemblance to the tutor and father's songs (if anything, their songs are more similar to the father's song (i.e., there seems to be one bird that is driving the lack of significance)). If this is indeed what the authors intend to communicate, they should revise the wording here to more accurately reflect the analyses. The same applies to the description of the analysis of syllable rate.

We appreciate the reviewer's suggestion to explicitly state the analysis and its corresponding results. We would like to highlight that in the previous revision round, our manuscript included a statistical analysis to directly compare the similarity of the pupil to playback with the similarity of the pupil to the genetic father. Following previous feedback, we re-analyzed the data using linear mixed-effects models incorporating the father-ID as a random effect.

To address both concerns, we have now decided to reword our manuscript to include both, the mixed-effects model and the two-sample t-test analysis, explicitly stating the comparisons and the corresponding statistical results (lines 158-171):

- I thank the authors for modifying the equation pertaining to syllable rate but can the authors include a justification for this in the manuscript? Even now the formula seems odd to me. It's basically the inverse of the average syllable duration (which was what was calculated in the previous submission), which is not the same as the canonical definition of syllable rate. For example, in Mets and Brainard 2019 (a paper that is cited by the authors ([44]), syllable rate ("tempo") is calculated as "the number of syllables present in a song bout divided by the duration of the song bout". In other words, this calculation includes the duration of gaps in the denominator to give a common measure of tempo (see also Leadbeater et al., 2005; Forstmeier et al., 2009; Rouse et al, 2023; etc.).

§ The year of Mets and Brainard is 2019, not 2017.

We appreciate Reviewer #1's comment and acknowledge that our measure of syllable rate is indeed the inverse of the average syllable duration. Given that gap duration was not affected by the experimental conditions, we opted to exclude any additional measurements that contribute to song duration. Therefore, we will not use the previous definition outlined in Mets & Brainard (2017, 2019). As we clearly define our measure in the Methods section and do not refer to it as "tempo", we respectfully believe that further justification is not necessary, as it is simply a matter of definition.

- Figure 2, line 202: There are no "dashed" lines in the figure, just black lines

Changed.

- Line 232: "female call ratio" is not defined before this, making it hard to understand. Please explicitly define.

Defined (lines 225-226).

- Legend for Figure 3D: indicate that this is the 95% CI (to harmonize with text).

Changed.

- Line 284-286: why use this statistic? They should still use a mixed effects model since there are multiple neurons per bird. The distribution of differences looks like it could be different than zero, depending on which neurons belonged to which birds. I would make the same point for statistics related to Figure 4L and all other data.

We replaced our statistic with a mixed effect model (outlined in the Supplementary Table1 and results highlighted in the text (lines 279-280) and the legend of Figure 4L).

- Line 294: delete “during song practice” because these juveniles are not singing.

Changed.

- Line 302-304: The authors can explicitly analyze the strength of responses to female calls outside or during BOS playback because 15 neurons were recorded under both conditions. So if the authors really want to make this statement, they should conduct the analysis. They should not base this statement on differences in the level of significance between panels C-F and G-L because sample sizes are larger in the analyses depicted in C-F (therefore more power).

Due to a potential difference in baseline activity evoked by bird’s own song alone, comparing the difference in response strength to these two different stimuli is problematic. We therefore changed the wording and only comment on the observed differences in the stereotypy of the inputs, meaning whether the addition of a female call induced correlated subthreshold activity (lines 295-296).

- Legend for Figure 4L: add SE or SD.

Data added (Figure legend 4L).

- Legend for Figure 4: Generally speaking, while I appreciate the details about the data, if the authors are going to add details about means in the figures for the BOS conditions (G-L), they should do the same for the non-BOS condition (C-F) for consistency. But honestly, I don’t think they need to report the means here.

Data is adjusted accordingly.

- Line 310: add “extracellular” here to be clear and to avoid other readers from wanting an explicit comparison between juvenile and adult data. Could also add to figure 5 to avoid confusion.

Changed.

- Figure 6G: I wouldn’t use “BOS” here since the same acronym is used to refer to HVC activity during the playback of the bird’s own song. Write “singing” or something that differentiate these data from the data in Figure 4.

Changed.

- Line 383: Figure 5 should not be referenced since this figure summarizes data from adults.

Changed.

- Line 418: "When female vocalizations were not correlated with the learning phase"... what is the learning phase? And what does it mean to not be correlated with it? Does it refer to when juveniles are practicing (singing), the developmental period of vocal practice, or something else?

Rephrased.

- Line 551: DAS was defined only in a figure legend, and TCN has not been defined so please write out both here.

Changed.

- Line 561-563: A little more clarity about the window of interest is important for people to replicate these experiments and analyses. Is the beginning of the window the onset of the first introductory note, and if so, what is the duration of silence used to differentiate if an introductory note is part of the song bout or not. Also, just to clarify, the end of the window is the offset of the last syllable of the bout 350 ms?

Clarified in lines 567-571.

- Line 572-575: please clarify if the exact same part of song is extracted for the control condition? Is that what "at the same duration from song onset as the perturbed snippets" is supposed to indicate? It seems unclear to me from the current wording.

Since the juvenile birds' songs are highly variable during the song learning phase, it wasn't always possible to find the exact matching syllable in the unperturbed versions. To address this, we extracted snippets from unperturbed songs using the same onset and offset timing as the perturbed songs. This process is now clarified in the methods section (lines 583-584).

- Supplementary Figure 1: It seems like this bird didn't learn the tutor song so I would just say it "produced a" song that....

Changed.

- Supplementary Figure 3: what does each data point represent? a single exemplar or the mean from a particular syllable type?

Clarified in the Figure legend of Supplementary Figure 3: Each data point represents a similarity score to tutor song of a single song snippet (either perturbed (orange) or unperturbed (green)).

REVIEWERS' COMMENTS

Reviewer #1 (Remarks to the Author):

I thank the authors for the additional analyses and details. This manuscript is much improved, and I appreciate the more conservative description of the key data on HVC responses to female calls during vocal practice. I don't have any major concerns but I have one comment related to statistical analysis and have some comments about wording. I look forward to seeing this research in print!

Line 43: add "of" ("subset OF HVC")

Figure 3 legend: when people plot 95% confidence intervals, there are lines on both sides of the mean. So, while I understand that the important comparison is of tutored birds to the upper CI, I suggest that the authors also plot the lower CI. Alternatively, the authors can rephrase the description of the dashed line in the legend.

Title for Figure 5 legend. To be consistent with the title of Figure 4, I suggest the title "Female vocalizations elicit neural responses in a subset of HVC projection neurons in adult birds."

Line 325-326 add "in a subset of"

Line 352-354: given that multiple neurons were analyzed per bird, this data should be analyzed using a mixed effects model.

Title for Figure 6 legend: To be consistent with the title for Figure 4, I suggest "Female vocalizations during juvenile song production changes neural activity within a subset of HVC projection neurons"

Figure 6 legend: change BOS to "singing"

Line 461-462: rephrase, it sounds like the authors removed (omitted) "auditory responses" from the dataset: I suggest "since HVC neurons in adult birds do not demonstrate auditory responses to female calls during song."

Line 584-586: redundant with previous sentence?

Supplementary figure 2: I just want the authors to confirm that there is NOT a significant negative relationship between age and song production in juveniles 1 & 2 (since "n.s." is indicated above each of these bird's plot). The data certainly seem compelling.

A point-by-point response to the reviewers' comments:

REVIEWERS' COMMENTS

Reviewer #1 (Remarks to the Author):

I thank the authors for the additional analyses and details. This manuscript is much improved, and I appreciate the more conservative description of the key data on HVC responses to female calls during vocal practice. I don't have any major concerns but I have one comment related to statistical analysis and have some comments about wording. I look forward to seeing this research in print!

We thank Reviewer #1 for their thoughtful feedback, constructive comments, and encouraging words regarding our manuscript's improvements and future publication. We incorporated the suggested comments as follows:

Line 43: add "of" ("subset OF HVC")

Added.

Figure 3 legend: when people plot 95% confidence intervals, there are lines on both sides of the mean. So, while I understand that the important comparison is of tutored birds to the upper CI, I suggest that the authors also plot the lower CI. Alternatively, the authors can rephrase the description of the dashed line in the legend.

The description of the dashed line in the Figure legend was rephrased.

Title for Figure 5 legend. To be consistent with the title of Figure 4, I suggest the title "Female vocalizations elicit neural responses in a subset of HVC projection neurons in adult birds."

Changed.

Line 325-326 add "in a subset of"

Changed.

Line 352-354: given that multiple neurons were analyzed per bird, this data should be analyzed using a mixed effects model.

In this case we compare categories of responses (number of spikes during female call perturbations in HVC of juvenile versus adult birds) and not continuous outcome variables in which case a linear mixed effect model would be more suitable. We therefore respectfully disagree with Reviewer #1 and leave the statistical assessment unchanged.

Title for Figure 6 legend: To be consistent with the title for Figure 4, I suggest "Female vocalizations during juvenile song production changes neural activity within a subset of HVC projection neurons"

Changed.

Figure 6 legend: change BOS to "singing"

Changed.

Line 461-462: rephrase, it sounds like the authors removed (omitted) "auditory responses" from the dataset: I suggest "since HVC neurons in adult birds do not demonstrate auditory responses to female calls during song."

Changed.

Line 584-586: redundant with previous sentence?

Removed.

Supplementary figure 2: I just want the authors to confirm that there is NOT a significant negative relationship between age and song production in juveniles 1 & 2 (since "n.s." is indicated above each of these bird's plot). The data certainly seem compelling.

We agree with Reviewer #1 that the data look compelling, but we did not find a significant relationship between age and song practice in juvenile 1 or 2.